



Atmospheric
Chemistry
and Physics

# Photochemical production of ozone and emissions of NO$_x$ and CH$_4$ in the San Joaquin Valley

**Justin F. Trousdell**[1], **Dani Caputi**[1], **Jeanelle Smoot**[2], **Stephen A. Conley**[3], **and Ian C. Faloona**[1]

[1]Department of Land, Air, and Water Resources, University of California Davis, Davis, CA, USA
[2]Department of Chemistry, University of California Davis, Davis, CA, USA
[3]Scientific Aviation Inc., Boulder, Colorado, USA

**Correspondence:** Ian C. Faloona (icfaloona@ucdavis.edu)

**Abstract.** Midday summertime flight data collected in the atmospheric boundary layer (ABL) of California's San Joaquin Valley (SJV) are used to investigate the scalar budgets of NO$_x$, O$_3$, and CH$_4$, in order to quantify the individual processes that control near-surface concentrations, yet are difficult to constrain from surface measurements alone: these include, most importantly, horizontal advection and entrainment mixing from above. The setting is a large mountain–valley system with a small aspect ratio, where topography and persistent temperature inversions impose strong restraints on ABL ventilation. In conjunction with the observed time rates of change this airborne budgeting technique enables us to deduce net photochemical ozone production rates and emission fluxes of NO$_x$ and CH$_4$. Estimated NO$_x$ emissions from our principal flight domain averaged 216 ($\pm$33) metric t d$^{-1}$ over six flights in July and August, which is nearly double the California government's NO$_x$ inventory for the surrounding three-county region. We consider several possibilities for this discrepancy, including the influence of wildfires, the temporal bias of the airborne sampling, instrumental interferences, and the recent hypothesis presented by Almaraz et al. (2018) of localized high soil NO emissions from intensive agricultural application of nitrogen fertilizers in the region and find the latter to be the most likely explanation. The methane emission average was 438 Gg yr$^{-1}$ ($\pm$143), which also exceeds the emissions inventory for the region by almost a factor of 2. Measured ABL ozone during the six afternoon flights averaged 74 ppb ($\sigma$ = 9.8 ppb). The average midafternoon ozone rise of 2.8 ppb h$^{-1}$ was found to be comprised of $-0.8$ ppb h$^{-1}$ due to horizontal advection of lower O$_3$ levels upwind, $-2.5$ ppb h$^{-1}$ from dry deposition loss, $-0.5$ ppb h$^{-1}$ from dilution by entrainment mixing, and 6.9 ppb h$^{-1}$ net in situ photochemical production. The O$_3$ production rates exhibited a dependence on NO$_2$ concentrations ($r^2 = 0.35$) and no discernible dependence on methane concentrations ($r^2 \sim 0.02$) that are correlated with many of the dominant volatile organic compounds in the region, suggesting that the ozone chemistry was predominantly NO$_x$-limited on the flight days. Additionally, in order to determine the heterogeneity of the different scalars, autocorrelation lengths were calculated for potential temperature (18 km), water vapor (18 km), ozone (30 km), methane (27 km), and NO$_x$ (28 km). The spatially diffuse patterns of CH$_4$ and NO$_x$ seem to imply a preponderance of broad areal sources rather than localized emissions from cities and/or highway traffic within the SJV.

## 1 Introduction

The setting for this research is the San Joaquin Valley (SJV) (see Fig. 1) which is the southern end of California's Central Valley, one of the largest valleys in the world by area. The SJV is a complex mesoscale environment where the surrounding topography limits the low-level inflow to the valley and makes vertical mixing particularly important to atmospheric boundary layer (ABL) ventilation, similar to the conditions in the Po Valley of Italy (Maurizi et al., 2013). Estimates of the coverage of mountainous terrain on the Earth's land surface varies anywhere from $\sim$ 25 % to 70 % (Grab, 2000; Noppel and Fiedler, 2002; Rotach et al., 2014), depending on the subjective criterion used, and thus orograph-

ically induced mesoscale circulations are of paramount importance in understanding the Earth–atmosphere exchange (EAE) over much of the continental land area. Horizontal inhomogeneities in the Earth's land surface affect the adjacent ABL in a variety of ways leading to pronounced changes in the EAE involving sea breezes (Miller et al., 2003), internal boundary layers (Garratt, 1990), and orographic effects (Rotach et al., 2015). Additionally, valleys are popular areas for human inhabitation due to lowland access, access to river waterways, and fertile soils for agriculture (Small and Cohen, 2004).

The SJV is well known for its persistent air quality challenges (Lagarias and Sylte, 1991; Cox et al., 2013). As of 2013 the Valley is a non-attainment site for the state (California Air Resources Board, CARB) and federal 8 h standard for $O_3$, a status that is only aggravated by the recent reduction of the federal 8 h standard to 70 ppbv (US EPA). Moreover, the majority of the SJV, especially its southern end, has been designated non-attainment for $PM_{2.5}$ under the state and federal standards since 2013. The need to understand and find solutions to these air quality issues has been the catalyst of numerous studies, including major multi-researcher field campaigns. In 1990, the San Joaquin Valley Air Quality Study (SJVAQS) was conducted. The largest study of its kind in the US at the time, the SJVAQS targeted the complexities of the SJV at a time when it was considered the nation's second worst overall air quality problem (Lagarias and Sylte, 1991). In 2000, the Central California Ozone Study (CCOS), a multi-year program of meteorological and air quality monitoring, emission inventory development, and air quality simulation modeling, held its intensive observation period. Finally, in 2010 the California Research at the Nexus of Air Quality and Climate Change Study (CALNEX) (Ryerson et al., 2013) was conducted across southern California and the SJV. These traditional studies tended to focus on ground-based atmospheric chemistry observations in the SJV, measuring as many different components of the oxidation chemical mechanism as possible in one location, for example at a "supersite" in Fresno, CA (Watson et al., 2000). However, a prominent meteorological process that can strongly influence surface concentrations is mesoscale advection by the horizontal flow, and due to the complexity of the surface wind field in complex terrain and the heterogeneity of surface sources this process's contribution to local air quality problems is difficult to account for in these types of studies. Furthermore, in studies that do deploy airborne platforms, the flight data tend to be limited in duration, overextended in sampling domain, and/or altogether uncoordinated with the surface sites, never attempting to measure advection quantitatively.

Another essential process influencing air quality at surface sites is mixing at the top of the ABL or entrainment. Entrainment, the dynamical process whereby a turbulent mixed layer incorporates adjacent fluid that is laminar or much less turbulent, predominantly drives the daytime ABL growth, and

is generally a diluting process when considering trace gases with surface sources (or precursors). Local ABL air affected by surface emissions is diluted with background, less turbulent, and typically warmer and dryer air in which pollutant concentrations remain relatively low (Stull, 1988). This classical image is complicated, however, when polluted air is transported locally, regionally, and/or synoptically to the free atmosphere above the ABL before being entrained, which is often the case in mountainous terrain.

Wheeler et al. (2010) investigated ozone events that occurred during CCOS and found that the tendency of photochemical models to underestimate peak ozone was likely due to an underrepresentation of emissions, particularly from wildfires, as well as regional recirculation and transport of ozone and/or ozone precursors aloft across the model's boundaries. Polluted ABL air can also be vertically recirculated in complex terrain by slope venting along valley sidewalls only to be reincorporated into the valley boundary layer via entrainment (Fast et al., 2012; Leukauf et al., 2016; Henne et al., 2004). Moreover, a growing body of evidence is suggests that distal air pollution can represent a significant source of local air quality degradation in the western US as a result of entraining air masses that have been transported across the Pacific (Parrish et al., 2010; Huang et al., 2010; Lin et al., 2012; Pfister et al., 2011; Ewing et al., 2010).

In addition to issues of long-range transport, mesoscale dynamics, and turbulent mixing, there are outstanding questions about the chemistry and sources of pollutants in the SJV. Pusede and Cohen (2012) suggested the existence of a temperature-dependent volatile organic compound (VOC) in the SJV and their results indicated that the trend in ozone exceedance days, at least over the past dozen years or so, was due to a transition to $NO_x$-limited photochemistry and ongoing $NO_x$ reduction strategies in the region. Even with a well-accepted theory of ozone chemistry, discrepancies still exist between measured and modeled ozone from regional air quality models (Brune et al., 2016). That study found ozone production rates from $HO_2$ around the morning rush hour to be double modeled rates when NO typically reached its highest diurnal levels, and measured $HO_2$ in instances of the very highest observed NO was seen to rise to more than 10 times the modeled values. Another study from the same CALNEX surface data set posited that an unknown temperature-dependent VOC, quite possibly of agricultural origins, dominates OH reactivity at high temperatures when $O_3$ problems are most likely (Pusede et al., 2014). Agricultural sources of $NO_x$, an important precursor for ozone production and a dangerous pollutant in and of itself, were estimated using three independent methods in the study of Almaraz et al. (2018), suggesting that California's croplands may account for 20 %–51 % of the state's overall $NO_x$ emissions, while current CARB inventories assume that the contribution from soils is insignificant.

The purpose of this study is to employ in situ aircraft data, including meteorological and chemical data, collected pre-

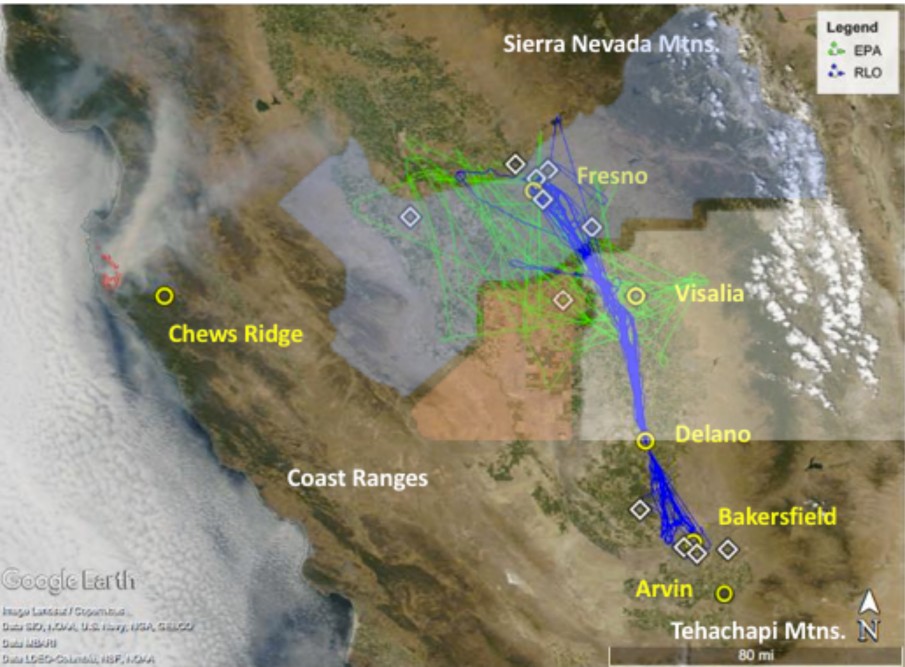

**Figure 1.** Satellite view of the San Joaquin Valley (© Google Earth 2019 TS1). Yellow circles are key sites. The green lines are the combined tracks from the EPA flights and the blue lines are from the RLO flights. White diamonds are CARB surface stations. Active fires on the ground can be seen as small red outlines to the northwest of the Chews Ridge site (available at: https://worldview.earthdata.nasa.gov/, Worldview, 2018) from 27 July 2016. Major mountain ranges are labeled in white. The three shaded regions are Fresno County in blue, Kings County in red, and Tulare County in white.

dominately during the summer of 2016 in the SJV for an integrated study of ozone, NO$_x$, and methane, employing a scalar budget technique. The individual terms of the scalar budgets, which are responsible for the observed overall time rates of change in the ABL, are calculated, enabling a relative comparison of each individual process. This method includes treatments of both horizontal advection and entrainment mixing – essential processes not well captured in modeling or ground-site studies.

## 2 Geophysical setting and the buffer layer

In the southern SJV prevailing northwesterly surface winds (parallel with the valley axis) slow down as they converge against a topographical cul-de-sac leading to stagnation. The SJV has a long and deep geography, running approximately 400 km (Stockton to Arvin), bounded at over 3 km on its northeastern flank (southern Sierra Nevada), $\sim 1$ km to its southwest (Diablo and Temblor ranges), and $\sim 2$ km at its terminus (San Emigdio and Tehachapi mountains; see Fig. 1). The surface airflow in the SJV comes through gaps and cools in the Pacific Coast Ranges, predominately around the San Francisco Bay Area and through the Carquinez Strait, bringing fresh emissions of NO$_x$ and VOC precursors from those urban areas. These precursors generate ozone concentrations that typically increase as the air mass moves south-

east, often reaching a maximum in the southern end of the valley near Bakersfield (Cox, 2013). This north to south gradient can be seen in Fig. 2 from four observation stations in the SJV, showing the annual pattern of the probability of ozone exceedances from 10 years of CARB data spanning 2006 to 2015. However, the horizontal distribution of ozone is not always so straightforward and different "background" meteorological conditions have been shown to distort this general pattern (Jin et al., 2011).

Elevated temperature inversions above the SJV in the summer, which are present almost every day of the year (Iacobellis et al., 2009), constrain vertical air motions and impede the venting of air pollution. These inversions, coupled with the topographic isolation of the SJV air along the valley floor, make the valley's air quality strongly dependent not only on local emissions but also on the nature of the entrainment mixing. The air above the ABL in the SJV is unique in that it does not purely consist of background air from the free troposphere (FT) as in most cases over flat terrain. A three-layer conceptual system has been presented in Faloona et al. (2019) for the SJV comprised of the ABL, a buffer layer, and the FT. This buffer layer is an admixture of background air masses aloft (FT), which flow over the Pacific Coast Ranges with a Froude number on the of order 0.1 and stagnate against the Sierra Nevada, mixed with SJV boundary layer air that is transported vertically along the val-

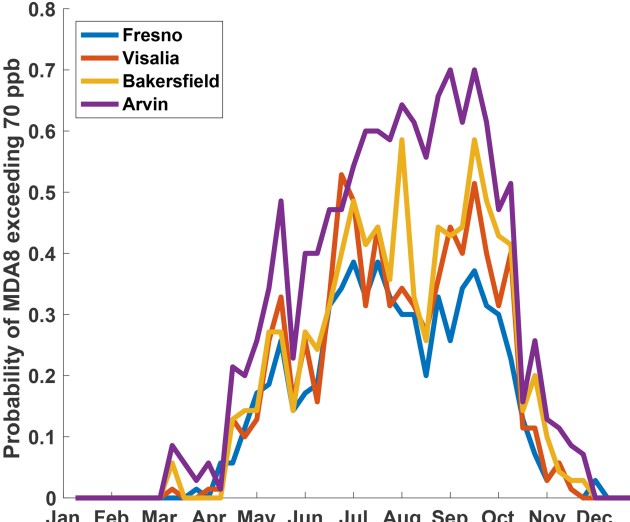

**Figure 2.** The probability of an MDA8 exceeding 70 ppb each week of an annual cycle at four sites in the SJV, from north to south: Fresno, Visalia, Bakersfield, and Arvin (see Fig. 1). The data are from the CARB network from 2006 to 2015.

ley sidewalls on its transit up the valley. The vertical extent of the buffer layer begins atop the ABL, which across the region during the summer in the afternoon average from around 600 m, up to roughly 2000 m (a.g.l.). The approximate residence time within this buffer layer was found to be about 1 week based on analysis of Weather Research and Forecasting (WRF) model output that we plan to detail in a forthcoming paper.

## 3   Methods

### 3.1   WRF model configuration

The Weather Research and Forecasting (WRF) model version 3.8.1 was used in hindcast to provide vertical velocities essential to the study but not measured by the aircraft. The model was configured using a pair of two-way nested domains initialized at 12 and 4 km resolution. Much of the coarser domain covers the western United States, while the finer-resolution domain is centered around California and Nevada. This model configuration features 50 terrain-following vertical levels, with 30 levels being located below 5 km in height, and an increased resolution near boundary layer heights within the SJV. The Moderate Resolution Imaging Spectroradiometer (MODIS) data set was used for land usage categories. The North American Regional Reanalysis (NARR) data set was used to initialize model runs, and new initial conditions were introduced every 3 h. In addition, four-dimensional data assimilation (FDDA) was utilized in the coarse domain for wind speeds in every vertical level, and temperature and water vapor within the lowest vertical

level and above the planetary boundary layer. FDDA used the National Centers for Environmental Prediction (NCEP) Administrative Data Processing (ADP) Global Surface Observational Weather Data (ds461.0) and Upper Air Observational Weather Data (ds351.0), both of which are at 6 h temporal resolution, to nudge model runs.

### 3.2   Aircraft instrumentation

Aircraft data were collected by a Mooney Bravo and Mooney Ovation, which are fixed-wing single-engine airplanes operated by Scientific Aviation Inc. The wings are modified to sample air through inlets under the starboard wing that draw air to the onboard analyzers through 3.18 mm (outer diameter) stainless steel (for $CH_4$, $CO_2$, and $H_2O$) and 6.35 mm PFA Teflon (for $NO_x$) and PCTFE (for $O_3$) tubing. Temperature and relative humidity were measured by a Vaisala HMP60 humidity and temperature probe. Ozone was measured with a dual-beam ozone absorption monitor (2B Technologies Model 205) with a nominal accuracy and precision for 10s averages of 2 % for concentrations above 50 ppbv. A Picarro wavelength-scanned cavity ring-down spectrometer (WS-CRDS) measures $CH_4$, $CO_2$, and water vapor (Picarro model 2301-f) with a specified 30 s precision of < 2 ppbv, < 200 ppbv, and < 150 ppmv, respectively. NO was measured by chemiluminescence (ECO PHYSICS Model CLD 88) with a stated precision of $\sim$ 2 %. A blue-light LED photolytic converter (model 42i BLC2-395, manufactured by Air Quality Design Inc.) was used to selectively convert $NO_2$ to NO for alternating measurements of $NO_x$ (=$NO + NO_2$). The instrument was cycled through the states of NO and total $NO_x$ every 20 s. Calibrations were performed by $O_3$ titration with a NIST(National Institute of Standards and Technology) traceable NO standard (Scott-Marrin Inc.), certified to within 5 %. Full calibrations were performed before and after the entire flight series, with zero and span checks run routinely before and after each flight. Additionally, every 10 min the sample flow and the instrument's generated ozone was redirected through a pre-reaction chamber for a 40 s period where the $NO+O_3$ reaction and subsequent chemiluminescence (CL) was allowed to take place before the detection cell, thereby tracking any matrix interferences that may add to the usual CL in flight. These background signals interpolated between the 10 min intervals were then subtracted from the continuous measurements. The interpolated $NO_2$ signal was noted to decay approximately exponentially after powering up, which sometimes affected the first 15–30 min of flight. The presumed artifact was successfully replicated in the lab with a constant $NO_2$ concentration and was removed by exponential detrending (see the Supplement). Winds are measured on the aircraft using a dual-hemisphere global positioning system combined with direct airspeed measurements, as described in Conley et al. (2014).

## 3.3 Flight strategies

The flights specifically target that time of the day when the ABL is actively growing but only after passing the original rapid growth phase through the neutrally stable residual layer (RL). The main data set we use here comes from six flights sponsored by the US EPA (labeled EPA in Fig. 1) during the California Baseline Ozone Transport Study (CABOTS) that were conducted on the afternoons of 26–28 July and 4–6 August 2016 from 11:00 to 15:00 PST, spanning an approximate altitude range from near the surface up to $\sim 3$ km (Supplement Fig. S1). The aircraft flights consisted of six or seven back-and-forth level and profiling legs of approximately 15 min duration ($\sim 60$ km), primarily along the mean wind direction (the valley axis) in order to capture the horizontal advection and vertical gradients of the measured scalars. The flight domain focused on the region of the SJV between Fresno and Visalia, with approximately two-thirds of the data collected below $\sim 1$ km and missed approaches executed at each airport in order to sample to within a few meters of the ground. The flight days were selected in coordination with a team from NOAA's Earth System Research Laboratory operating a tunable optical profiler for aerosol and ozone (TOPAZ) lidar in Visalia, California, who have shown excellent agreement between the ozone data collected by the aircraft and lidar (Langford et al., 2019). Periodically the airplane would make deep vertical profiles from $\sim 3$ m to 3 km in addition to two or three other profiling legs in order to diagnose the ABL top, its growth, and vertical profiles of the measured scalars.

Another 15 flights were flown as part of a residual layer ozone study (from now on referred to as RLO flights) with a different flight pattern from the previous 6 (EPA) flights mentioned. The afternoon RLO flights were shorter in duration, did not cover the cross-valley dimension significantly, and consisted of direct transects from Fresno to Bakersfield and back with approximately six vertical profiles over approximately 2.5 h between 12:30 and 15:00 PST. Aside from take-offs and landings at Fresno, these flights also included five very low passes at the Visalia, Delano, and Bakersfield airports (yellow circles in Fig. 1) in order to sample within $\sim 5$ m of the surface. Despite their shorter duration and elongated domain, we include some analyses from these flights because they are more numerous and offer valuable complementary information to this study. Details of the RLO flights can be found in Caputi et al. (2019).

## 3.4 Scalar budgeting technique

The quantification and categorization of the essential processes that determine the surface concentrations of these pollutants can be executed by targeted flight campaigns. Such experiments provide a valuable service to the air quality community, especially modelers interested in testing their models on a process basis. After quantifying the individual terms of the budget equations, each term's relative importance can be weighted to provide a better understanding of the leading causes and factors affecting surface concentrations. Outlined in the seminal work of Lenschow et al. (1981) are the original applications of the scalar budgeting techniques used by Warner and Telford (1965) and Lenschow (1970) to help validate the newly developing technique of eddy covariance for measuring sensible heat fluxes by aircraft. Lenschow et al. (1981) went on to describe the effectiveness of well-designed aircraft ABL studies in determining the net source or sink (in their case for ozone) given the careful measurement of the other dynamically controlled terms. The technique can be generalized to any scalar budget (i.e., ozone, NO$_x$, water vapor, dimethyl sulfide (DMS), SO$_2$) to enable the calculation of important residuals, including source or sink terms for non-conserved species (Bandy et al., 2011; Conley et al., 2009; Faloona et al., 2009; Kawa and Pearson, 1989). Boundary layer heights were determined from each profile (approximately 8–12 per flight) based on the abrupt increase in potential temperature and drop in water vapor. The locations and time of each of these observations were then fit by a multilinear regression in time and the horizontal dimension to determine the ABL growth rates and gradients, which go into the budget to determine the entrainment velocity (Trousdell et al., 2016). Taking all the airborne data observed below the derived (linear) time-dependent ABL depth, we then perform the same multi-linear regression for all the scalars, including potential temperature, water vapor, O$_3$, NO$_x$, and CH$_4$. Aligning the $x$ axis with the mean wind direction, $U$, the advection and temporal trend terms of Eq. (1) are derived from the coefficients of the linear regression fit to the ABL NO$_x$ concentration field in horizontal direction and time (Conley et al., 2011). For a more in-depth discussion of the airborne budgeting technique and specifics for the budgets of methane and ozone in the SJV, see Trousdell et al. (2016). The calculation of our emission estimates necessitates that we find an effective area of the ground that encompasses all the sources that have influenced the ABL air mass that is sampled. For each of the six flights we simply drew a polygon enclosing the latitude and longitude coordinates of the aircraft sampling within a time-dependent ABL, whose height was parameterized using the linear regression mentioned above. The average area of this polygon was 5200 km$^2$ ($\sigma = 940$ km$^2$). To estimate an uncertainty in this area, we consider the average advection distance of the mean wind ($\sim 3$ m s$^{-1}$) over the course of a large eddy turnover time (boundary layer height divided by convective velocity scale, which was $\sim 7$ min $= 550$ m/1.4 m s$^{-1}$) and multiply this on either end of the domain by an average cross-valley dimension (70 km) to generate a "spread" in the sampled ABL area influenced by the surface flux field. Although this additional area represents less than 4 % of the overall domain, we include a conservative 20 % error in the error analysis for it. The $\sigma$ total flight time in the SJV was 22 h, including 8 h in the ABL.

### 3.4.1 $NO_x$ budget

Calculating the budget for $NO_x$ requires evaluating the terms in the following equation:

$$\frac{\partial [NO_x]}{\partial t} = \frac{F_0}{z_i} + \frac{w_e \Delta [NO_x]}{z_i} - \frac{\overline{[NO_x]}}{\tau_{NO_x}} - U \frac{\partial [NO_x]}{\partial x}. \quad (1)$$

The budget terms are (in order from left to right) a storage term $\left(\frac{\partial NO_x}{\partial t}\right)$; the difference between the surface flux ($F_0$) and entrainment flux, which is comprised of the entrainment velocity ($w_e$) and the jump in $NO_x$ concentration ($\Delta [NO_x]$) across the entrainment zone divided by ABL height ($z_i$); the chemical loss term, which is the mean $NO_x$ concentration ($[NO_x]$) divided by the photochemical lifetime of $NO_x$ ($\tau_{NO_x}$); and horizontal advection ($-U \frac{\partial NO_x}{\partial x}$). Unlike our other budgets, calculating the $NO_x$ budget requires estimating the loss term because of its short photochemical lifetime. The oxidation rate of $NO_x$ is principally controlled in the daytime by reaction with OH. Therefore, the rate constant $k_{NO_2+OH}$ was estimated from the equation and data presented for termolecular reactions given by JPL (Burkholder, 2015), with an average temperature and pressure taken from our flight data (average effective first-order reaction rate, $k_{NO_2+OH}[M] \sim 1.0 \times 10^{-11}$ cm$^3$ molec$^{-1}$ s$^{-1}$). The median midday peak OH we chose to use in our calculation was observed in a different study to be approximately $6$–$8 \times 10^6$ molec cm$^{-3}$ in the San Joaquin Valley (Brune et al., 2016), with a flight duration average of about $6 \times 10^6$ molec cm$^{-3}$, which yields an average afternoon $NO_x$ photochemical lifetime, $\tau_{NO_x}$, of $\sim 4.6$ ($\pm 0.08$) h for the six flights.

### 3.4.2 Error analysis

The error for each derivative term in our multilinear regressions is a root-mean-square error (RMSE) derived in the fit. The entrainment fluxes are comprised of the entrainment velocity and a scalar delta term. The delta term error was assigned to be $-1.0$ ppb for $NO_x$ and $-50$ ppb for methane, based on variations in the data estimated by eye from inspection of the many vertical profiles. The entrainment velocity estimates are obtained from (see Supplement Sect. S1) derivatives of ABL height, whose errors were previously mentioned, and a term from the WRF model (subsidence or vertical velocity), which we have estimated as a conservative $0.5$ cm s$^{-1}$, and the horizontal wind at ABL height assigned an error of $0.2$ m s$^{-1}$ based on the measurement capabilities of the instrument (Conley et al. 2014). The same error for horizontal winds near ABL height applies to the ABL horizontal winds used in calculating the advection terms. The $NO_x$ equation has a chemical loss term with an error from the uncertainty estimate equation for termolecular reactions, given by JPL (Burkholder, 2015) and the error in averaged ABL $[NO_2]$, employed in the chemical loss term, was taken

as a standard deviation of all the measurements. Estimated emission terms are residual terms within the respective budget equations. Their errors are calculated by adding the relative errors of all the other terms in the budget in quadrature. The regional area used to scale up the emission flux was assigned an error of $20\%$. The error in our average emission rates for $NO_x$ and $CH_4$ for all of the flights is a standard error of the mean (the standard deviation of the estimates divided by $\sqrt{6}$). We believe that the errors in our emission estimate on a given flight day are likely much larger than any actual day-to-day variability, so the repeated flight dates amount to multiple measurements of a value that is approximately constant; therefore, we consider it appropriate to treat the reported error of regional emissions as the standard deviation of the mean. In other words, while the emission estimate from a single flight on any given day is quite a noisy measurement, the repeated experiment can build confidence in the average result.

## 4 Results and discussion

In the following section we present a variety of inferences gleaned from the three scalar budgets performed for $NO_x$ to derive regional surface emissions (Sect. 4.1.1), for $O_3$ to derive afternoon photochemical production rates (Sect. 4.1.2) and to see how that fits into the overall diurnal budget of ozone (Sect. 4.1.2), and finally for $CH_4$ to derive regional emissions (Sect. 4.1.3). Because of the large discrepancy between our estimates of $NO_x$ emissions and that of the state inventory, we further explore possible reasons to explain the difference. The first is the hypothesis put forward by Almaraz et al. (2018) that there is a substantial source of NO from fertilized agricultural soils that is not accounted for in current state inventories (Sect. 4.1.1). The second is the possibility that the Soberanes fire in the mountains of the Coast Range approximately 200 km to the west (see Fig. 1) may have influenced our $NO_x$ budget in the ABL around Fresno (Sect. 4.1.1). The third explores the bias introduced by measuring only during the afternoon when $NO_x$ emissions are thought to be highest (Sect. 4.1.1) and the fourth discusses the possibility of a chemical interference in the measurement of $NO_2$, which in our system relies on photolysis followed by the chemiluminescence measurement of NO (Sect. 4.1.1). The interference hypothesis is further explored by calculating Leighton ratios (Sect. 4.1.1) in order to determine if the observed $NO_2 : NO$ ratios appear consistent with the theoretical photostationary state between $O_3$, NO, and $NO_2$ expressed in the Leighton ratios. This latter point leads naturally to the discussion of our estimates of ozone photochemical production (Sect. 4.1.2) because it is, in principle, related to deviations in the observed Leighton ratios. Next, we present the observed spatial patterns of these scalars in the ABL, calculating their horizontal autocorrelation lengths (Sect. 4.2) to potentially infer emissions heterogeneity.

## 4.1 Budgets

In a future companion paper we will present the surface sensible heat fluxes for our flight region via two independent methods. The first being a turbulence analysis of the horizontal ABL winds using mixed-layer similarity considerations, the second a scalar budget analysis – in exactly the same manner as what is done for chemical species in this work – for potential temperature in the ABL, and finally a comparison of these to the output of the land surface parameterization of the WRF model. The results support each other and afford us added assurance in the budgeting technique. Supporting information from 15 additional flights in the SJV are presented for a project focused on studying ozone over the diurnal cycle, with a focus on residual layer ozone, and used for some of the analysis (RLO flights). In general, we expect the EPA flights to yield better results because they were longer in duration and more geographically focused targeting a complete midday budget of the scalars. Nevertheless, we performed the same analysis on the midday RLO flights and there do appear to be significant differences between the two domains (see Fig. 1) when looking at the averaged quantities between the campaigns. For example, the entrainment rates for the entire region down to the southern end of the SJV at Bakersfield are nearly 50 % larger than those around Fresno and Visalia. This is an interesting finding and one that is consistent with generally deeper boundary layers found in the southern end of the SJV, as pointed out in previous studies (Bianco et al., 2011; Trousdell et al., 2016).

### 4.1.1 $NO_x$ emissions

$NO_x$ ABL data were filtered by eliminating data greater than a standard deviation above the mean before being analyzed in order to remove the skewness from the distributions induced by numerous spikes encountered in the late afternoon. Variations in this threshold from 1 to 3 standard deviations did not change the mean flight concentration by more than 2 %–3 % so the exact threshold was not considered critical for our analysis. The data filtering was done to eliminate the spikes, which were consistently encountered throughout the latter part of the flights, each lasting no more than a few minutes and uncorrelated with any other species measured ($CO_2$, $CH_4$, and $O_3$). We suggest that their source may have been something in the fire smoke entrained in the late afternoon ABL that caused a transient interference in the $NO_2$ photolytic chamber (they were not observed in the NO measurements). Furthermore, as we discuss later in conjunction with Fig. 5, the influence of the fires on $NO_x$ measured by the surface network ($\sim 1$ ppbv) appears to be minimal during the six EPA flights relative to the clear signal enhancements in CO ($\sim 50$ ppbv) and $PM_{2.5}$ ($\sim 10 \mu gm^{-3}$). Because the spikes were only encountered in the later afternoon their influence was particularly troublesome in estimates of the secular trend in $NO_x$. In four cases, simply removing the spikes from the ABL data set permitted a reasonable estimate of the afternoon trend, but on two flights we resorted to using data from the CARB monitoring network (https://www.arb.ca.gov/adam/hourly/hourly1.php, last access: TS2 November 2018). The trend established was the average of three station trends (from 11:00 to 16:00 PST) throughout the region: Fresno (Garland), Visalia (N. Church St.), and Hanford (S. Irwin St., the white diamond in Fig. 1 just west of Visalia). The estimates from the surface network and aircraft were very comparable for the other four flights where both were measured (averages of $-0.38$ vs. $-0.34$ ppb h$^{-1}$, respectively).

Results from the $NO_x$ budgeting are shown in Table 1. The average $-0.36$ ppb h$^{-1}$ secular trend of $NO_x$ is similar in magnitude to dilution induced by entrainment ($-0.30$ ppb h$^{-1}$), with chemical loss ($-1.4$ ppb h$^{-1}$) and surface emission ($1.3$ ppb h$^{-1}$) approximately balancing out. On average, advection is not significant but on any given day (for instance, 29 July) it can be large, which has been found to be the case elsewhere for other scalars and geophysical settings (Conley et al., 2009, 2011; Faloona et al., 2009). The relative role of entrainment changes from day to day. When compared to the average chemical loss, it is about 15 %, but on two of the flights (27 and 28 July) it is almost double that relative magnitude.

Measured emissions for the flight region were averaged and converted to metric tons per day giving an estimate of 216 metric t d$^{-1}$ ($\pm 33$, standard error; see error analysis Sect. 3.4.2). The California Air Resources Board's (CARB) total $NO_x$ inventory, representative of the summer based on CARB's CEPAM 2016 SIP Standard Emission Tool (available at: https://www.arb.ca.gov/app/emsinv/fcemssumcat/fcemssumcat2016.php, CEPAM 2018), is 103.7 metric t d$^{-1}$ for the three surrounding counties: Tulare, Fresno, and Kings in SJV (Fig. 1). The combined size of these three counties is about 6 times the flight region area. Thus, we would expect the airborne sampling domain to be a subset of the three-county region; however, since $\sim 86$ % of the $NO_x$ sources in the CARB inventory are mobile for these counties and our sampling occurred in the vicinity of each county's major population center (Visalia, Fresno, and Hanford) and one of the SJV's main traffic arteries (CA state Highway 99), it may be reasonable to expect the countywide $NO_x$ emissions to be mostly sampled by the flights. Calculated $NO_x$ chemical lifetimes averaged out to be 4.62 h ($\sigma = 0.08$) for all flights, based on the rate constant data for nitrogen dioxide's reaction with the hydroxyl radical to form nitric acid. We discuss several possible explanations for the discrepancy in our emission estimate and those of the CARB inventory in the following sections: soil $NO_x$ emissions from fertilized agriculture in the region, wildfire effluent impacts on the airborne measurements, the bias due to the daytime sampling times, and possible chemical interferences in the measurement.

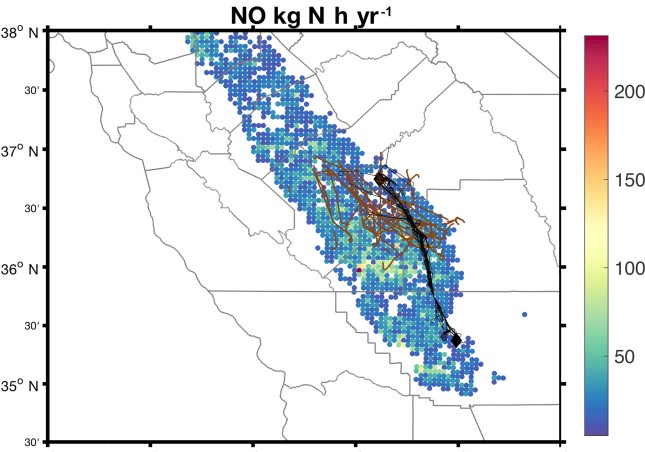

**Figure 3.** Soil NO emissions in kilograms of nitrogen per hectare per year for the SJV with flight tracks (data from Almaraz et al., 2018). Significant sources for soil NO show up in the middle SJV in Kings County.

**Table 1.** $NO_x$ budgets for the six EPA flights. All averages are calculated using ABL data only. The columns labeled $1\sigma$ represent the estimated error in the preceding term with the same units.

| Flight date | $\partial[NO_x]/\partial t$ storage (ppbv h⁻¹) | $1\sigma$ | $-U(\partial[NO_x]/\partial x)$ advection (ppbv h⁻¹) | $1\sigma$ | $w_e\Delta[NO_x]/z_i$ entrainment (ppbv h⁻¹) | $1\sigma$ | $-k[OH][NO_2]$ chem loss (ppbv h⁻¹) | $1\sigma$ | $F_0/z_i$ emission (ppbv h⁻¹) | $1\sigma$ | $\tau NO_2$ lifetime (h) | $1\sigma$ | Avg. ABL $[NO_x]$ (ppbv) | $1\sigma$ | Avg. ABL $[NO]$ (ppbv) | $1\sigma$ | Regional emissions (t d⁻¹) | $1\sigma$ |
|---|---|---|---|---|---|---|---|---|---|---|---|---|---|---|---|---|---|---|
| 27/07/2016 | −0.29 | 0.04 | 0.01 | 0.03 | −0.52 | 0.37 | −1.5 | 1.1 | 1.7 | 1.1 | 4.68 | 1.46 | 9.0 | 3.5 | 1.9 | 3.6 | 274 | 187 |
| 28/07/2016 | −0.09 | 0.07 | −0.09 | 0.03 | −0.52 | 0.39 | −1.4 | 0.9 | 1.9 | 1.0 | 4.68 | 1.45 | 8.5 | 2.6 | 2.1 | 3.0 | 301 | 166 |
| 29/07/2016 | −0.44 | 0.07 | 0.30 | 0.05 | −0.17 | 0.14 | −1.8 | 2.1 | 1.2 | 2.1 | 4.69 | 1.47 | 9.6 | 8.8 | 1.4 | 7.9 | 182 | 324 |
| 04/08/2016 | −0.73 | 0.01 | 0.04 | 0.01 | −0.30 | 0.21 | −1.1 | 0.7 | 0.6 | 0.7 | 4.60 | 1.40 | 5.5 | 1.6 | 0.5 | 1.2 | 115 | 132 |
| 05/08/2016 | −0.83 | 0.03 | −0.01 | 0.01 | −0.15 | 0.13 | −1.5 | 1.0 | 0.9 | 1.0 | 4.51 | 1.36 | 7.8 | 2.9 | 0.8 | 2.3 | 137 | 160 |
| 06/08/2016 | 0.22 | 0.06 | −0.02 | 0.01 | −0.14 | 0.13 | −1.3 | 0.8 | 1.7 | 0.9 | 4.55 | 1.37 | 7.8 | 2.4 | 1.8 | 3.8 | 286 | 155 |
| Average | −0.36 | 0.05 | 0.04 | 0.02 | −0.30 | 0.23 | −1.4 | 1.1 | 1.3 | 1.1 | 4.62 | 1.42 | 8.0 | 3.6 | 1.4 | 3.7 | 216 | 187 |
| SD | 0.4 | 0.02 | 0.1 | 0.01 | 0.2 | 0.1 | 0.2 | 0.5 | 0.5 | 0.5 | 0.08 | 0.05 | 1.4 | 2.6 | 0.6 | 2.3 | 81 | 69 |

## Soil $NO_x$ emissions from agriculture

CARB currently considers mobile sources to make up 86.3 % of the total $NO_x$ emissions and that agriculture contributions are negligibly small. Nonetheless, agriculture represents the largest source of nitrogen to the state in the form of synthetic fertilizers (32 %) and animal feed (12 %), with about half of what is being applied to crops being lost to the environment (Tomich, 2016). Parrish et al. (2017) studied the temporal change in the ozone design values for California air basins over the past 3 decades and three heavily agricultural regions stood out: San Joaquin Valley, Salton Sea (containing the Imperial Valley), and the northern Central Coast (containing the Salinas Valley). Parrish et al. (2017) went on to fit the trends of the air basins to that of the South Coast Air Basin in their mathematical model in order to optimize their parameters, but in doing so had to leave out the data in the SJV before 2000. From 1980 to 2000 the trend essentially plateaus for the SJV, and since 2000 the trend is anomalously slow in the Salinas Valley and has an uncommonly high offset in the Salton Sea. The authors go on to suggest that these irregularities in the decadal $O_3$ trends may be explained by agricultural emissions and/or by some unspecified temperature-dependent VOC with a possible connection to agricultural practices as proposed by Pusede and Cohen (2012).

While the average $NO_x$ surface concentration decreased in the SJV by about 9.3 % over the years 2005–2008, the Sacramento, San Francisco, and South Coast regions saw a range between 22.6 and 30 % decrease (Russell et al., 2010). In addition, the modeling estimates of Almaraz et al. (2018) show concentrated regions of high NO emissions in the SJV (Fig. 3), Salton Sea air basin, and the Salinas Valley with the greatest magnitude $NO_x$ emissions from soils for the state in these areas. In their model for soil $NO_x$, temperature was

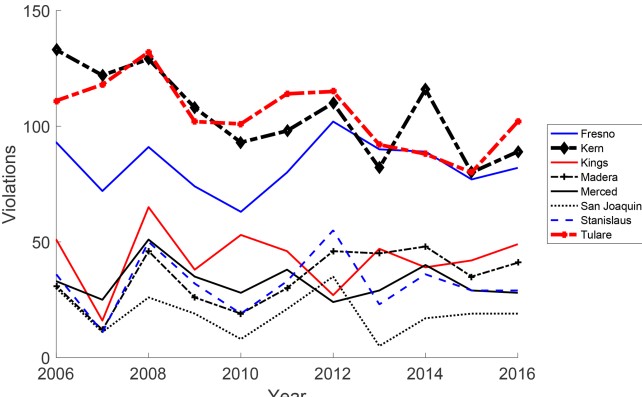

**Figure 4.** Tulare and Kern counties show signs of a downward trend for ozone violations, while the other counties of the SJV, which are largely rural, do not show a clear downward trend. Los Angeles County is shown as a reference for comparison with the South Coast Basin. Data provided by California Air Resources Board.

tracked, as well as water-filled pore space and nitrogen availability. Following a sensitivity analysis, they found temperature to be one of the primary factors influencing soil $NO_x$ emissions in the presence of excessive application of fertilizers where soil microbial communities increase the availability of nitrogen. We found a weak correlation ($r^2 = 0.18$) between our emission estimates and the ABL potential temperature, which should be just a little lower than the surface air temperature so we consider it as a decent proxy for soil temperature. Looking at the number of violations of the maximum 8 h daily average $O_3$ (MDA8) standard for the counties of the SJV (data provided by CARB) over the past decade indicates no observable trends outside of Kern and Tulare counties, which contain some of the larger urban centers: Bakersfield, Visalia, and Hanford (Fig. 4). While the SJV air basin as a whole may be showing slight decreases in MDA8 $O_3$ standard violations much of its rural areas are not. In a satellite study, Russell et al. (2010) point out that changes to the spatial extent of $NO_2$ in the SJV are slower than other regions of the state. Other regions with stronger urban influences show significant shrinkage of the average $NO_2$ cloud around major urban centers while the SJV is largely an amorphous cloud of $NO_2$ in the satellite images. Nevertheless, two other counties with major urban centers: Fresno County containing the city of Fresno, and San Joaquin County, containing the city of Stockton, do not show decreasing trends in the max 8 h daily ozone yearly trend since 2006. Pusede and Cohen (2012) present satellite data from 2007 to 2010 that show a significant $NO_2$ cloud around the Stockton area and, to a lesser extent, around Fresno, although the SJV as a whole shows more broadscale homogeneity than other regions in California.

## Potential influence from wildfires

It is important to note that throughout the course of the EPA flights the Soberanes fire was burning along the Big Sur coast of California. The fire started on 22 July 2016 and lasted until October of the same year. From the NASA MODIS satellite, clear images can be seen of the fire smoke being advected out and above the valley ABL on some days (see Fig. 1). We found greater variability towards the end of the flights in the ABL $NO_x$ data, which could possibly be explained by the entrainment of fire smoke as the ABL reaches its maximum height. Amongst the myriad chemical emissions from wildland forest fires is $NO_x$ (Urbanski et al., 2009), and globally Jaegle et al. (2005) estimate that biomass burning contributes $\sim 14\%$ of surface [$NO_x$]. Singh et al. (2012) sampled numerous fire plumes throughout California during 2008 and found very little $NO_x$ ($< 0.5$ ppb) near the source of the fires but that the plumes could later acquire $NO_x$ by mixing with other air masses containing higher $NO_x$ levels. When the fire plumes they measured mixed with substantially polluted urban air, ozone formation rates were found to be at their highest all across California in comparison to purely urban or rural air. Elevated levels of reactive nitrogen oxides ($NO_y$) have been observed in smoke from biomass burning that contains reservoir species for $NO_x$ like peroxyacetyl nitrate (PAN) that can later release $NO_x$ relevant to $O_3$ formation (Dreessen et al., 2016).

Taking CARB data from their Fresno–Garland surface site, which falls within our flight region, we looked at CO, $PM_{2.5}$ and $NO_x$ for the 6 EPA flight days. While we found a strong positive correlation between CO and $PM_{2.5}$, we found no correlation between CO and $NO_x$. In a longer time series representative of the Soberanes fire (22 July–12 October 2016) a small, positive, yet weak, correlation between CO and $NO_x$ can be seen (see Fig. 5). Because the leading term of chemical loss is directly proportional to $NO_2$ concentration in the $NO_x$ budget equation (Eq. 1), a sensitivity test was run to see how changes in the $NO_2$ concentration affect the emission estimates. A change in 1 ppb of $NO_2$ on average changes the emission by 35 metric $t\,d^{-1}$. Nevertheless, even though there was likely some influence of the fire on the regional $NO_x$ levels, the contribution most likely entered the ABL top through entrainment, which in principle is accounted for in the budgeting method by changes in the average jump across the ABL ($\Delta NO_x$).

Diurnal CARB surface data from the SJV were compared to $NO_x$ measurements from the RLO flight data below 100 m and at night we see fairly good agreement between the two. During our afternoon ABL flights, however, there appears to be a positive bias in the flight data (Fig. 6). This may be because the airborne data below 100 m (a.g.l.) altitude were only able to be collected during the low approaches, take-offs, and landings at the airports: Bakersfield, Delano, Fresno and Visalia, all of which are located in close proximity to CA State Highway 99.

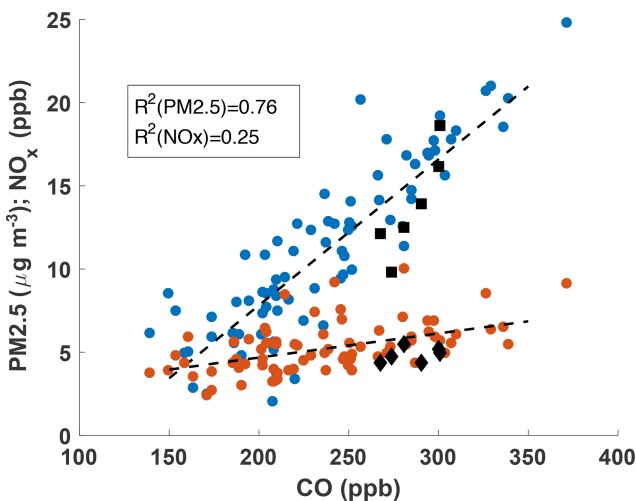

**Figure 5.** Data from CARB for Fresno–Garland and Clovis sites during Soberanes fire (22 July–12 October 2016). Here we see a correlation between CO and $PM_{2.5}$ (blue dots) but a weaker correlation between CO and $NO_x$ (orange dots). The black squares ($PM_{2.5}$) and diamonds ($NO_x$) are the six EPA flight dates.

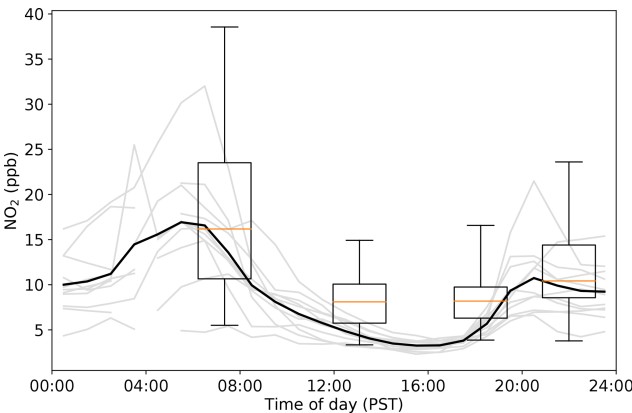

**Figure 6.** Box and whisker plots for flight data from RLO flights; the grey lines are data from 11 CARB surface sites (see Fig. 1). Statistics for the box and whisker plot (lower adjacent, 25th percentile, median, 75th percentile, upper adjacent): sunrise (−2.0, 10.7, 16.2, 23.5, 42.7), afternoon (−0.3, 5.7, 8.1, 10.1, 16.5), evening (2.6, 6.3, 8.2, 9.7, 15), and night (0.6, 8.6, 10.4, 14.4, 23.1).

**Potential bias because of the measurement period**

Considering a typical diurnal cycle of $NO_x$ emissions in a region with a large urban influence, particularly traffic, our emission rate during the period of measurement can be roughly estimated to be about a factor of 1.4 greater than the average emission rate over the entire diurnal cycle. This is based on work by de Foy (2018), who estimated diurnal profiles of emissions based on a model which took into consideration the timing of mobile sources and meteorological impacts on their concentration measurements in the Chicago area (see their Fig. 9 modeled data). Furthermore, Russell et

al. (2010) reported a 27 % decrease in $NO_2$ concentrations on weekends for Fresno and Bakersfield. Because five of our six flights were on weekdays and the last flight was a Saturday, our sampling may be slightly biased toward weekdays. Assuming an average decrease in $NO_x$ emissions on weekends to 0.73 the weekday rate, our average daily emission rate would be a factor of 1.04 (= $(5.73/6.46) \times (7/6)$) higher than inventories, which average over the 5 weekdays and 2 weekend days each week. Taken together, the timing of the flights relative to the inventory's average summer emission rate could lead to a positive bias in our measurements of $\sim 45\%$ (= $1.4 \times 1.04$).

**Possible chemical interference in $NO_x$ measurements**

The photolytic converter setup was developed specifically to avoid the problem of converting nitrogen-containing species other than $NO_2$, which has been well established for standard CL $NO_x$ monitors employing a heated molybdenum catalyst (Dunlea et al., 2007). Reed et al. (2016) studied interferences in the photolytic $NO_2$ instruments and found PAN to be the most significant due to thermal decomposition in the lamp chamber, reporting that in their instrumental setup $\sim 5\%$ of the PAN is dissociated. Although this is likely a very small contribution in our study, where average ABL temperatures were $\sim 305$ K, there is a very real possibility of some compounds that decompose to $NO_2$ (e.g., peroxynitrates, $RO_2NO_2$, or possibly even alkyl nitrates, $RONO_2$) being present in large quantities in the wildfire plumes (Alvarado et al., 2010; Akagi et al., 2011) that we encountered over the valley on some days. The short-lived spikes discussed earlier were removed from our analysis, and even when included they did not significantly affect the reported average $NO_x$ concentrations. Consequently, these transient interferences should not impact our estimates of regional $NO_x$ emission rates. However, the time-dependent interference that was removed (see the Supplement), could be the result of some wildfire effluent compound coating the inside of the photolysis cell and contributing to the aggregate to the average $NO_x$ measurements. Inspection of the collection of all $NO_x$ interferences observed in the field and post-field calibrations and zero tests, which were removed in the final data analysis, show that the largest impacts were around 2 ppbv $NO_x$ in magnitude. Using the sensitivity to emission estimates we calculated and discussed above, the very largest imaginable uncorrected interference in our $NO_x$ measurements could give rise to an overestimate in our emissions of $\sim 70$ t d$^{-1}$, reducing our result by about 32 %.

**The Leighton ratio**

Here we use a modified Leighton ratio, $\Phi'$, to estimate the possible range of interference to our measured $NO_2$. The Leighton ratio, $\Phi$, is unity when the $NO_x$ and ozone chemistry is in photostationary state (PSS) and there is no net

ozone photochemical production. The ratio deviates above unity when some other chemical process produces NO$_2$; i.e., when reactions involving peroxy radicals augment the primary pathway of NO oxidation by O$_3$. This is usually associated with areas where NO$_x$ concentrations are not too high, like heavily polluted urban centers where there are greater sinks for peroxy radicals and the loss pathway of [OH] with NO$_2$ is significant (Cantrell et al., 1993; Volz-Thomas et al., 2003). Deviations below unity indicate possible strong local NO emissions or rapid changes in $j_{NO_2}$, so that PSS has not been reached (Ma et al., 2017). Equation (5) from Griffin et al. (2007) defines a modified ratio, $\Phi'$, TS3 where it is assumed that peroxy radicals (RO$_x \equiv$ RO$_2$ + HO$_2$) alone are responsible for deviations seen in the Leighton ratio:

$$\Phi = \frac{j_{NO_2}[NO_2]}{k_1[NO][O_3]}, \Phi' = \frac{j_{NO_2}[NO_2]}{k_1[NO][O_3] + k_2[NO][RO_x]}. \quad (2)$$

Forcing the modified ratio to be unity we can solve for NO$_2$: reaction rates come from JPL kinetics data, the photolysis rate is from the NCAR Quick-TUV calculator (available at: http://cprm.acom.ucar.edu/Models/TUV/Interactive_TUV/, Quick TUV Calculator, 2018), [O$_3$] is from our flight data, and [RO$_x$] is taken from measurements made near Bakersfield presented by Brune et al. (2016), taken as their [HO$_2^*$], which includes some amount of [RO$_2$] interference ([HO$_2^*$] = 15 pptv, during our flight times). Between the NO$_2$ concentrations observed on the EPA flights and the calculated NO$_2$ there is a correlation of $R^2 = 0.50$ and a difference in mean NO$_2$ of about 0.7 ppbv (6.6 for EPA flights and 5.9 ppbv calculated). This can be considered a conservative estimate for possible NO$_2$ measurement interference because the choice for [RO$_x$] is on the lower end of a range of possible values. Griffith et al. (2016) reported maximum measured values of HO$_2^*$ from about 3 to 40 pptv from Pasadena, California, in the summer of 2010 with a corresponding NO$_2$ range of about 6–14 ppbv (approximate values are taken from the timeframe of our flights for comparison). As stated above, the sensitivity of our emission estimate to changes in NO$_2$ is 35 t d$^{-1}$ for every 1 ppb change to NO$_2$; therefore, possible chemical interference accounts for 24.5 t d$^{-1}$ or a systematic error of +11 %. From our EPA flights, we found a range of $\Phi$ values from 1 to 3.3 with an average of 1.87. For measurement conditions similar to ours (predominately rural) the reported values for $\Phi$ are between 1 and 3 (Cantrell et al., 1993; Volz-Thomas et al., 2003; Mannschreck et al., 2004); therefore, we feel that the NO$_2$ measurements reported herein are not likely to be subject to interferences much greater than $\sim 10\,\%$.

### 4.1.2 Ozone photochemical production

The ABL averaged ozone was 74 ppb ($\sigma = 9.8$ ppb) from our flight data, which is close to the summertime surface network average for that region. Looking at the CARB sites from Fresno, Tulare, and Kings counties within and close to

our flight region and averaging over the flight hours (12:00–16:00 PST) for the summer months (JJA), the average ozone concentration was 70 ppb ($\sigma = 13$ ppb). The averaged ozone photochemical production was 6.3 ppb h$^{-1}$ ($\pm 3.3$) compared to rates found for the southern SJV to be between 4.1 and 14.2 ppb h$^{-1}$ in summer (Trousdell et al., 2016). Kleinman et al. (2002) modeled ozone production rates using observed data for five major US metropolitan areas and found median values ranging from 3.5 to 11.3 ppb h$^{-1}$.

The ozone budget breakdown is shown in Table 2. The VOC chemistry in the SJV is dependent on temperature. At moderate temperatures ozone production is VOC-limited, while at higher temperatures it is less so, according to work by Pusede and Cohen (2012). They speculate that the temperature-independent part of the organic reactivity in the southern SJV has been decreasing over the past few years in response to emission regulations and is what led to the sharp decrease in ozone exceedances from the mid-1990s until 2010 (Pusede et al., 2014). They further propose that NO$_x$ regulations will be the most effective way to reduce ozone exceedances in the future, and as NO$_x$ levels decrease the temperature-dependent aspect of ozone chemistry will be diminished because at higher temperatures it becomes even more NO$_x$-limited. Trousdell et al. (2016) also argued that the ozone production in their study from 11 flights south of Bakersfield in 2013–2014 was NO$_x$-limited based on their estimates of the VOC : NO$_x$ ratio derived from their airborne measurements of CH$_4$ as a VOC proxy and the surface network observation of NO$_x$. Another study by Brune et al. (2016) suggests that ozone production continues to increase with NO beyond about 1 ppb, in contrast to how the "weekend effect" is believed to occur; however, high values of NO mostly occurred in the Brune et al. (2016) study during the early morning before the timeframe of the Pusede et al. (2014) study (10:00–14:00 PST). The 'weekend effect' is a phenomenon where ozone production goes up on weekends as NO$_x$ emissions decrease due to less motor vehicle traffic on the roads, particularly heavy duty diesel trucks. Marr and Harley (2002), using an air quality model with a customized motor vehicle emissions inventory on 4 days in August 1990, reported a 30 % reduction of NO$_x$ emissions on the weekends in Central California but noted that the response of the ozone concentrations to the emissions reduction differed throughout the modeling domain with slight decreases in the area of the SJV included in the model (from Stockton south to Fresno). Russell et al. (2010) stated a 27 % decrease in NO$_x$ emissions on weekends for Fresno and Bakersfield, the largest cities in the SJV, from satellite data taken from the summers of 2005–2008. When we included the 15 additional (RLO) flights we found photochemical production rates of 7.2 ($\pm 4.0$) ppb h$^{-1}$ on weekdays (total of 15 flights), 7.8 ($\pm 2.4$) ppb h$^{-1}$ on the weekends (total of 6 flights), and a correlation with NO$_x$ concentrations ($r^2 = 0.35$, Fig. 7), suggesting that NO$_x$-limited conditions prevail. The average NO$_x$ concentrations were 8.5 ($\pm 2.0$) ppb on weekdays and

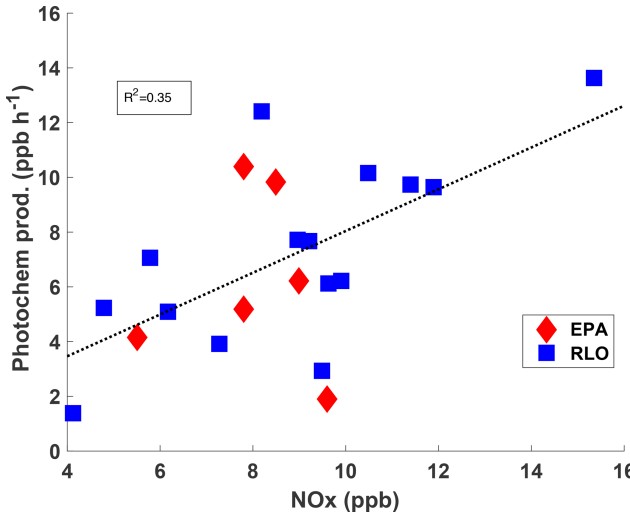

**Figure 7.** Correlation between photochemical production and $NO_x$ improved after including the additional 15 RLO flights. The correlation suggests that the flight region is in the $NO_x$-limited regime.

9.0 ($\pm$2.1) ppb on the weekend. No significant weekend effect was observed in this data set but one possible cause is that of the weekend flight days only 2 were Sundays, where the most significant $NO_x$ reductions are seen, and Saturday acts more like a transition day (Russell et al., 2010).

Assuming that $RO_x$ is responsible for positive deviations from PSS, we can, in principle, relate our ozone photochemical production rates to expected $RO_x$ levels. Assuming that net ozone photochemical production is solely due to $RO_x$ and making the simplifying assumptions that the reaction rates with NO of $RO_2$ is approximately equal to that of $HO_2$ (Mihelcic et al., 2003), we can estimate the gross photochemical production rate of ozone to be as follows:

$$P(O_3) = k_{NO+HO_2} \cdot [RO_x][NO]. \tag{3}$$

The reaction rate is for the reaction of NO with $HO_2$ (Burkholder, 2015). A similar approach is found in Mihelcic et al. (2003), who used it for calculating what they saw as an upper limit for $P(O_3)$, as well as Ma et al. (2017). Therefore, applying our own calculated net production rates and subtracting an estimated photochemical loss of $-1$ ppb h$^{-1}$ (due to photolysis and OH production, similar to the values Pusede et al., 2014, reported from their observations: 0.7–1.4 ppb h$^{-1}$) to get a gross production rate we expect the values for $RO_x$ to be a lower limit. Our results indicate a range of values 2.4–19.4 pptv with an average of 10.2 pptv. Brune et al. (2016) show afternoon values of about 8 pptv $HO_2$ and 15 pptv $HO_2^*$ (including some $RO_2$ interference) in the SJV, which is consistent with our findings because the airborne data are representative of a wide regional average. Our measurements are distinct from those made at the Bakersfield supersite during CALNEX, which is at the heart of an urban plume.

Next, looking at our modified Leighton ratios, $\varphi_1$, and using our measured concentrations with the JPL rate constants and solving for $RO_x$ we find an average value of 154 pptv. Assuming that this value is off by a factor of 3 as found by Mannschreck et al. (2004) this suggests an approximate average range for $RO_x$ during our measurement period of 9–50 pptv, and is consistent with several past studies (Cantrell et al., 1993; Hauglustaine et al., 1998 TS4; Volz-Thomas et al., 2003; Handisides et al., 2003) that found deviations in the Leighton ratio cannot be explained solely by the reaction of $RO_x$ with NO.

**Full diurnal budget of ozone**

Data from the SJV (Trousdell et al., 2016) indicate that $O_3$ production generally increases as you progress southward, as expected because of the predominant wind direction in the valley and the gradual accumulation of ozone precursors as the air mass moves up-valley (Cox, 2013). Like our budget equations, the prognostic equations of a State Implementation Plan (SIP) model track the different rate and derivative terms that sum to the total time derivative of any scalar of interest like ozone. Thus, it is important to know how these change over time, so here we present a diurnal analysis for our flight dates (Fig. 8). Ozone data are taken from eight CARB sites within our flight region, and our average photochemical production rate is extrapolated across the daytime hours by scaling the average observed value during the flight interval throughout the rest of the day based on the time series of $J(O^1D)$ from the NCAR Quick-TUV calculator. The areas under the curves represent the total [$O_3$]; therefore, it can be seen that the contributions from photochemistry and mixing down from the RL (fumigation) are approximately comparable. Very similar fifty-fifty split contributions from these two terms have been presented by past studies (Kleinman et al., 1994; Lin, 2008; Neu et al., 1994).

Now, we present an analysis of the average diurnal cycle of $d[O_3]/dt$ and $d[O_x]/dt$ ($O_x = NO_2O_3$, used because in the morning $NO_2$ quickly photolyzes to form $O_3$) (Fig. 9) for the SSJV based on our RLO aircraft observations. Average trends are taken across each $\sim$ 2 h flight, as well as estimated in between flights, totalling eight estimates each day. Data are binned into three altitude layers: the lower boundary layer 0–200 m – within the nocturnal boundary layer, as discussed by Caputi et al. (2019); the upper boundary layer 200–600 m (mainly the RL when it is present and within the afternoon ABL); and the buffer layer 600–2000 m. Looking at the near-surface data from the RLO flights and Fig. 8 we see similarities: a peak (dominated by fumigation from the RL) around 08:00–09:00 PST, a zero crossing around 15:00 PST, and a max loss at 19:00 PST. The exact timing of these events may differ by about an hour or so, and we observed such discrepancies even between diurnal profiles from different CARB sites in the SJV (data not shown). Looking at the difference between the Bakersfield and Fresno sites we see a smaller

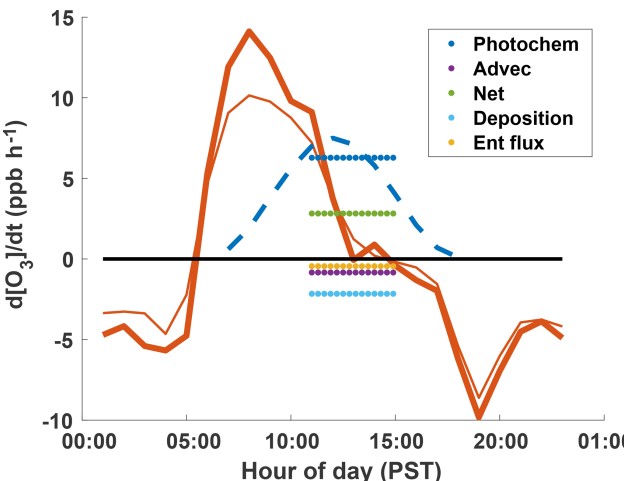

**Figure 8.** The thick orange line comes from CARB data from seven sites in the vicinity of the EPA flights (see Fig. 1) for June, July, and August. The thinner orange line is from the 6 flight days themselves. Horizontal dotted lines show each respective averaged ozone budget term over the flight hours and the EPA flights. The rapid increase in ozone levels in the early morning corresponds to when the ABL entrains residual layer air. This figure helps in visualizing the breakdown of the O$_3$ budget during the flight hours in comparison to the time rate of change of O$_3$ observed by local surface stations.

peak an hour later at Fresno and a smaller yet slightly earlier max loss time than at Bakersfield. To get a better sense of the peak loss rate, we compare the same results from the d[O$_x$] / d$t$ (Fig. 9), which shows that the near-surface drop in O$_3$ right after sunset is not simply due to titration by rush hour NO emissions because a very comparable loss is observed in O$_x$ that is conserved under titration. Given the absence of both photochemistry and entrainment at dusk, we conclude that large loss of O$_3$ must be occurring due to dry deposition in a severely stunted mixed layer. A more extensive analysis is needed to understand the variations in diurnal profiles of ozone production and loss across the SJV, but we propose it can be an instructive exercise to focus on the time derivatives as we have done here.

### 4.1.3 Methane emissions

The methane emission average, after conversion to more commonly reported units, was 438 gigagrams per year ($\pm143$, standard error) or 50 Mg h$^{-1}$ ($\pm15.5$), which is approximately one-half the size of an estimate by Cui et al. (2017) that used inverse modeling from flight data with a CALGEM (California Greenhouse Gas Emissions Measurement) prior and found about 80 Mg h$^{-1}$ ($\pm17$) for a region they labeled D1 in which our flights took place. Their D1 region contained Kern, Tulare, Madera, Fresno, and Kings counties, totalling about 58 billion square meters. The area used for calculating our emission here is the same area used in the NO$_x$ calculation. Our flight region was about 1/10 the

**Table 2.** TS5 O$_3$ budgets for the six EPA flights. All averages are calculated using ABL data only. The columns labeled 1$\sigma$ represent the estimated error in the preceding term with the same units.

| Flight date | $\partial[O_3]/\partial t$ storage (ppbv h$^{-1}$) | 1$\sigma$ | $-U(\partial[O_3]/\partial x)$ advection (ppbv h$^{-1}$) | 1$\sigma$ | $-vd[O_3]$ dry deposition (ppbv h$^{-1}$) | 1$\sigma$ | $w_e\Delta[O_3]/z_i$ entrainment (ppbv h$^{-1}$) | 1$\sigma$ | $\Delta[O_3]$ jump (ppbv) | Avg. ABL [O$_3$] (ppbv) | $P_{net}(O_3)$ photo. prod. (ppbv h$^{-1}$) | 1$\sigma$ |
|---|---|---|---|---|---|---|---|---|---|---|---|---|
| 27/07/2016 | 1.24 | 0.12 | −0.84 | 0.18 | −2.98 | 1.59 | −1.74 | 0.58 | −5.0 | 89.6 | 6.80 | 1.71 |
| 28/07/2016 | 6.06 | 0.19 | −2.35 | 0.13 | −2.47 | 1.32 | 1.04 | 0.48 | 3.0 | 70.2 | 9.83 | 1.43 |
| 29/07/2016 | −0.79 | 0.13 | 0.18 | 0.10 | −2.81 | 1.52 | −0.55 | 0.33 | −5.0 | 76.9 | 2.40 | 1.56 |
| 04/08/2016 | 0.95 | 0.05 | −0.48 | 0.09 | −2.11 | 1.11 | −0.99 | 0.32 | −5.0 | 75.7 | 4.53 | 1.16 |
| 05/08/2016 | 6.45 | 0.05 | −1.61 | 0.09 | −2.11 | 1.13 | −0.51 | 0.28 | −5.0 | 59.9 | 10.69 | 1.17 |
| 06/08/2016 | 3.00 | 0.05 | 0.04 | 0.05 | −2.33 | 1.24 | −0.47 | 0.29 | −5.0 | 70.9 | 5.77 | 1.28 |
| Average | 2.8 | 0.1 | −1.0 | 0.1 | −2.5 | 1.3 | −0.5 | 0.4 | −3.4 | 74.5 | 6.9 | 1.4 |
| SD | 3.3 | | 1.0 | | 0.4 | | 1.0 | | 3.6 | 10.8 | 3.5 | |

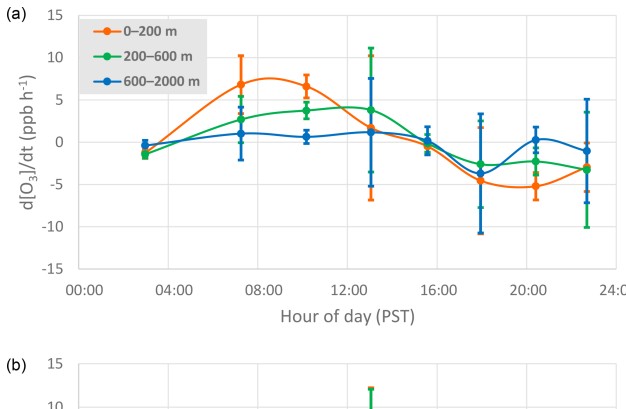

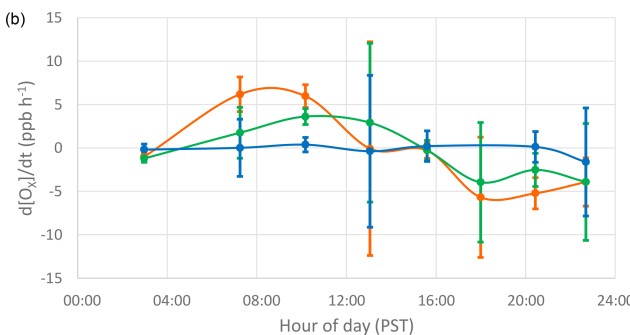

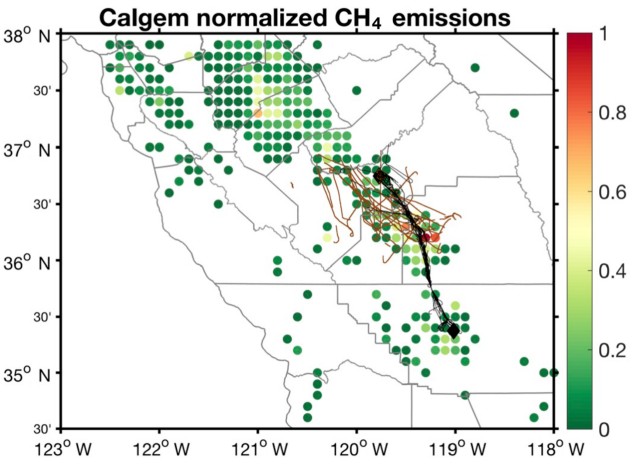

**Figure 10.** Normalized CALGEM inventory emissions showing the distribution and relative magnitude of methane sources with flight tracks (EPA flights in brown and RLO flights in black) superimposed. At the northern and southern ends of the RLO flights are black diamonds representing Fresno and Bakersfield, respectively.

**Figure 9.** The average time rate of change of **(a)** $O_3$ and **(b)** $O_x$ over a complete diurnal cycle. The curves are cubic polynomial interpolations between averaged data points from the RLO flights over each 2 h measurement period and intervals in between flights. Data are binned into three altitude layers: the lower boundary layer 0–200 m, the upper boundary layer 200–600 m, and the buffer layer 600–2000 m. Vertical bars around each point represent a standard deviation of the 15 flight sequences.

size of the D1 region, but the highest emission rates found in the Cui et al. (2017) study came from the region between Hanover and Visalia, on which our flights focused. Figure 10 shows our flight tracks overlaying the CALGEM emission inventories across the SJV. See Table 3 for a breakdown of methane budget terms. A region of the SJV of approximately 3.5 billion square meters was probed in a previous campaign from June through September 2013 and in June 2014 focused on the southern end of the SJV, particularly Bakersfield, reporting a measurement of 170 gigagrams per year ($\pm 125$) (Trousdell et al., 2016). Jeong et al. (2016), similar to Cui et al. (2017) and based on their CALGEM prior model, found that 86 % of the methane in the SJV is from dairies. Our flight area included one of two extremely dense areas of dairy operations in the valley focused around the intersection of three counties: Kings, Tulare and Fresno. Looking at the CALGEM inventory for our flight areas we found the average source apportionment for dairies to be 88 %. Trousdell et al. (2016) used CALGEM emission inventories scaled to the 2013 total $CH_4$ emission estimate for the state of California of 41.1 Tg $CO_2$ eq. provided by CARB and then compared that to in situ data and found top-down airborne to bottom-up inventory ratios of 3.6 and 2.4 for the two re-

gions studied (Fresno wintertime and Bakersfield summertime, respectively.) Our current study found an overestimate by a factor of 2.2 for the EPA flights. The site in Trousdell et al. (2016) with a value of 3.6 is dominated by emissions from petroleum operations (54 %), based on the CALGEM inventory, and took place during the winter while the other values mentioned come from regions dominated by dairy operations during summer. Cui et al. (2017) reported a ratio between their model inversion and CALGEM of 1.8, comparable to the value reported here and taking place in a similar region and season. Because the airborne measurements of our study take place during 6 summer days, comparisons with annually averaged inventories should be done with caution as methane emissions from the dominant sources (dairies) are likely to be seasonally temperature dependent. For example, a recent study of two dairies in the SJV (Arndt et al., 2018) reported facility-wide winter emissions to be only 40 %–50 % of those during summer sampling. If this very local study were applied to the annual variations throughout the region, it could, in principle, bring our estimates as well as those of Cui et al. (2017) in much closer congruence to the CALGEM inventory. More routine flights and applications of the budgeting methods outlined in this work could help further elucidate the accuracy of methane inventories.

## 4.2 Autocorrelation length scales

Autocorrelation lengths, or integral length scales, represent the distance over which a variable maintains a significant level of correlation with itself, or the minimum distance for which the variable becomes statistically independent (Tortell, 2005). Qualitatively, we think of this as the "patchiness" of the scalar field for our purposes in the horizontal dimen-

**Table 3.** $CH_4$ budgets for the six EPA flights. The columns labeled $1\sigma$ represent the estimated error in the preceding term with the same units.

| Flight date | $\partial[CH_4]/\partial t$ storage (ppbv h$^{-1}$) | $1\sigma$ | $-U(\partial[CH_4]/\partial x)$ advection (ppbv h$^{-1}$) | $1\sigma$ | $w_e\,\Delta[CH_4]/z_i$ entrainment (ppbv h$^{-1}$) | $1\sigma$ | emission rate (Gg yr$^{-1}$) | $1\sigma$ | $\Delta[CH_4]$ jump (ppbv) | Avg. ABL [$CH_4$] (ppbv) |
|---|---|---|---|---|---|---|---|---|---|---|
| 27/07/2016 | −40.5 | 1.89 | −3.4 | 2.20 | −69.4 | 22.1 | 487 | 362 | −200 | 2170 |
| 28/07/2016 | −8.0 | 0.91 | −6.9 | 0.70 | −69.5 | 24.6 | 982 | 447 | −200 | 2027 |
| 29/07/2016 | −10.5 | 1.32 | 9.3 | 0.92 | −22.1 | 12.8 | 31 | 177 | −200 | 2021 |
| 04/08/2016 | −5.7 | 0.58 | −3.4 | 0.21 | −39.7 | 12.7 | 686 | 290 | −200 | 2022 |
| 05/08/2016 | −5.3 | 0.45 | 0.0 | 0.24 | −20.5 | 10.8 | 223 | 170 | −200 | 1993 |
| 06/08/2016 | −3.1 | 0.49 | 1.6 | 0.26 | −18.9 | 11.6 | 220 | 190 | −200 | 1996 |
| Average | −12.2 | 0.9 | −0.5 | 0.8 | −40.0 | 15.8 | 438 | 273 | −200 | 2038 |
| SD | 14.1 | | 5.6 | | 24.0 | | 352 | | | 66 |
| SE | | | | | | | 144 | | | |

sions. Correlation coefficients were calculated as a function of distance by using a spatial autocorrelation technique called Moran's I:

$$CC = \frac{N}{W}\frac{\sum_i\sum_j w_{ij}(x_i-\overline{x})(x_j-\overline{x})}{\sum_i(x_i-\overline{x})^2}, \qquad (4)$$

where $w_{ij}$ is a weight matrix, which is either zero or one if the points of a pair $(i, j)$ are a certain distance from each other; $W$ TS6 is the number of pairs that fall into that distance category; $N$ is the total number of pairs in the data set; and $x$ represents the scalar. For our purposes, this means that all pairs of distinct scalar measurements in our domain are created and then binned into discrete bins based on distance between the two points that make up the pair. Then for each distance category a correlation coefficient is calculated and the bin width chosen was 1000 m. All data were selected to be within the time-dependent ABL and corrected to a common time and height within the ABL to remove biasing from temporal and vertical trends before the autocorrelation was run. The length scale was selected as the first crossing of the 0.37 line ($= e^{-1}$). The results averaged over the flights are: potential temperature (18 km), water vapor (18 km), ozone (30 km), methane (27 km), and $NO_x$ (28 km). Figures 11 and 12 show the horizontal distributions of the chemical species and the Moran's I decorrelation pattern for the flight of 27 July. Temperature and water represent ABL scalars dominated by surface fluxes, so in principle their correlation lengths are related to the scale of heterogeneity in their surface sources (irrigated or fallow fields, plots of differing albedos, urban heat islands, etc.) In the case of ozone, photochemical production dominates in the afternoon requiring the mixing together of $NO_x$ and VOC emissions. The more spatially diffuse pattern of $CH_4$ and $NO_x$, comparable to that of ozone, may imply a preponderance of broad areal sources rather than localized emissions from cities (5–15 km) and/or highway traffic. This result for $NO_x$ calls to mind the findings from Russell et al. (2010) and Pusede and Co-

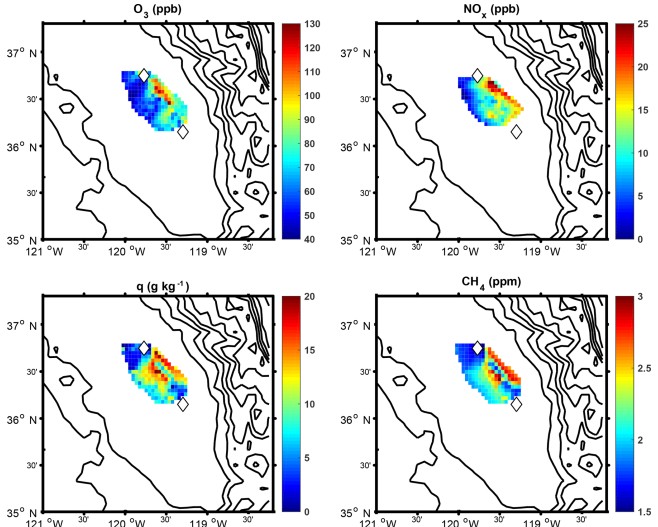

**Figure 11.** Scalar patchiness plots derived from ABL flight data corrected to a common time and height stamp with linear interpolation between data points within the flight domain. The white diamonds are Fresno and Visalia to the north and south of the area, respectively. Data taken from 27 July 2016.

hen (2012) that we have previously mentioned, which show broadscale homogeneity for $NO_x$ concentrations in the SJV unlike in other regions of California where urban hotspots appear more localized.

## 5   Conclusion

Using 6 d of flight data covering the period of ABL growth during the afternoon, we have estimated emissions estimates for $NO_x$ and $CH_4$ and photochemical production rates of ozone while employing a budget that exposes the key processes affecting their ABL concentrations. Of particular interest are the advection terms, which are very difficult to ob-

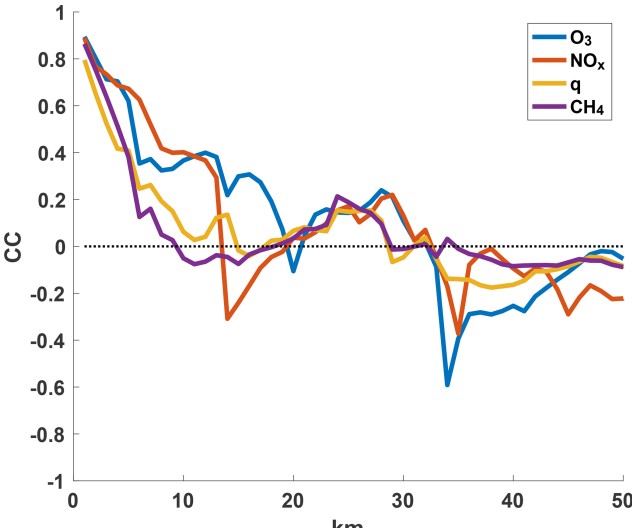

**Figure 12.** Spatial correlation plot corresponding to the patchiness plot from 27 July 2016. The $y$ axis shows the correlation coefficient (CC) and the $x$ axis shows the distance in kilometers for which the data have been correlated. $q$ signifies water vapor.

tain in ground-based studies, and entrainment, which is often not treated explicitly in models as it is fundamentally a turbulent, sub-grid process. Our emissions estimate for $NO_x$ suggest, like other previous studies, that agriculture in the SJV may be a greater source of $NO_x$ than previously thought and contributing more to the delayed decrease in $O_3$ surface concentrations compared to other air basins in California. After exploring possible explanations for $NO_x$ emission estimates being larger than expected, including a potential 45 % increase due to the afternoon (mostly weekday) timing of the flights and possible chemical interference accounting for at the very most 2 ppb to our average $NO_2$, we propose $79\,\mathrm{t\,d^{-1}}$ as the lowest conceivable estimate possible, after combining all of our conservative error estimates. With that said, our result is still significant because our study region accounts for only a fraction of the respective source region of the inventory estimate. Therefore, more work needs to be done to investigate soil $NO_x$ in the SJV as it offers a potential avenue for further air quality remediation by more efficient fertilizer usage in the valley. Emissions estimates from CALGEM for methane are possibly under-predicted by about one-half of the actual level for the SJV, in close agreement to other studies, but seasonality remains an important understudied factor. Finally, we suggest that calculations of autocorrelation lengths for $NO_x$, $CH_4$, water vapor, and ozone should be employed in future satellite and airborne studies in order to better understand the heterogeneity of sources in the ABL.

*Data availability.* All of the aircraft data used in this analysis can be found at https://www.esrl.noaa.gov/csd/groups/csd3/measurements/cabots/ (Caputi and Faloona, 2016; last access: 7

August 2019). WRF output can be requested from Ian Faloona, and CARB ground network data can be accessed from the CARB website (https://ww2.arb.ca.gov/, ARB, 2018; last access: 27 March 2018).

*Supplement.* The supplement related to this article is available online at: https://doi.org/10.5194/acp-19-1-2019-supplement.

*Author contributions.* JFT participated in the conceptualization, formal analysis, visualization, and writing of the manuscript. DC and JS participated in data collection, analysis, and visualizations for the work. ICF took part in conceptualization, funding acquisition, resources, methodologies, oversight of the project, and a significant portion of the writing. SAC was responsible for flying the aircraft and collecting the in situ data.

*Competing interests.* The authors declare that they have no conflict of interest.

*Acknowledgements.* We would like to thank Scott Bohning and Saffet Tanrikulu for their support. The work benefited from the coincident support of the California Air Resources Board. Ian C. Faloona's effort was supported by the USDA National Institute of Food and Agriculture (hatch project CA-D-LAW-2229-H, "Improving Our Understanding of California's Background Air Quality and Near-Surface Meteorology"). We thank the two anonymous reviewers for their thorough and critical reviews of this paper, which helped clarify the text significantly.

*Financial support.* This research has been supported by the Bay Area Air Quality Management District (grant no. 2016.129), the California Air Resources Board (grant no. 14-308), and the USDA National Institute of Food and Agriculture (grant no. CA-D-LAW-2229-H).

*Review statement.* This paper was edited by Robert McLaren and reviewed by two anonymous referees.

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

TS5     Storage was moved as requested.
TS6     Please confirm that it was fine to switch W and N or clarify.
TS7     Please add day.
TS8     11 was the issue number.
TS9     Please add day.