# Peer review of "Photochemical Production of Ozone and Emissions of NOx and CH4 in the San Joaquin Valley"

_Atmospheric Chemistry and Physics, 2018_

## Referee Comment (RC1) · Anonymous Referee #1 · 19 Feb 2019

The manuscript describes a number of flights over the San Joaquin Valley which measured NOx, O3, and CH4. These measurements are used to estimate surface emissions. For NOx and CH4, the surface emissions are much higher than inventory values.

General comments:

1) There are numerous typos and grammatical errors. Some of these are listed below, but I would suggest a thorough proof-reading before resubmitting the manuscript. Some of the language is also vague and inappropriate for a journal paper (e.g. "stuck out", "more or less", "more and more", "a lot").

2) No details of the flights are presented, with the exception of two sentences in Section 3.3. Figure 1 gives some idea of the horizontal extent of the two flight campaigns, although it is difficult to see the EPA flight lines and it is impossible to distinguish individual flights. We are given some windows of time (but no actual flight durations) and an altitude range "from the surface up to 4 km". Presumably the surface measurements are at the start and end of each flight, unless there were multiple landings at different airports. Given the importance of the vertical coverage on the emission calculation, much more detail is required.

3) The description of the NOx processing (Section 4.1.1) lacks detail. Why one standard deviation? How much data are removed? If this all occurs in the late afternoon, why not just reject data from that time of day? And if fire smoke is entrained during that time period, why is this effect not discussed for O3 and CH4?

4) If emission rates are measured over the course of a few hours for 6 days, then presenting these values as month or yearly rates (i.e. tonnes/year) is not "averaging" or "converting" - It is extrapolating. For the most part this can be fixed by using the correct terminology. However, due to variations in emissions with time (some of which are discussed, such as the "weekend effect"), extrapolating these hourly time scale values to yearly values has an associated uncertainty that should be discussed.

5) The consideration of uncertainties is generally weak. For example, in Section 3.4 (page 8, line 16) it is simply stated that the approach is justified and that 20% in conservative without any reference to where that number comes from. In Section 4.4, two values (1 ppb, 50 ppb) are chosen "because the term is estimated by eye". For wind, 0.1 m/s is based on the "measurement capabilities" of an instrument which isn't named or referenced.

6) Correlation length is typically calculated at the 1/e value, not the "crossing of the zero-correlation line". This is primarily because small amount of noise in the correlation values can significantly change when the zero line is first crossed (Figure 12 demonstrates this effect). Smoothing the correlation or fitting an exponential decay to

determine the value at 1/e gives a more accurate measure of the correlation length that isn't subject to the effects of noisy data.

7) It is also not clear why correlation distances are important in the context of the study (expect to inform future satellite resolution values). What is the expected correlation length that would be associated with cities and traffic? Wouldn't this value depend on how far downwind from the source the measurement is (due to horizontal diffusion)? Why are potential temperature and water vapor compared? Do these values relate to the patchiness of land use and the separation of lakes and rivers? Why is this important?

Specific comments:

Page 3, line 9-10: A claim like this is meaningless without defining "limited in duration", "overextended in sampling", and "altogether uncoordinated". If those terms can be defined a citation will also be needed to a substantial review paper that backs up this claim.

Page 5, line 17: For the editor – Is a citation of a manuscript in preparation accepted?

Page 7, line 11: The aircraft doesn't measure "from the surface".

Page 9, line 14: Using WRF parameterizations isn't a measurement.

Page 20, line 12-13: This sentence is very confusing. (e.g. What is "a common time height stamp"?)

Minor comments/corrections:

Page 2, line 23. Sentence doesn't make sense.

Page 3, line 14. Period should be outside bracket.

Page 5, line 3. What does "its" refer to?

Page 5, line 11. Why is the air "unique"?

Page 6, line 13. Should be "Vaisala". "Ozone" should start new sentence.

Page 9, line 22. Should be "50%".

Page 12, line 13. This is not a sentence.

––––––––––––––––––––––––––

---

## Referee Comment (RC2) · Anonymous Referee #2 · 8 Mar 2019

Trousdell et al. present data from flights over the San Joaquin Valley in California. They calculate NOx and CH4 emission rates for this region, as well as photochemical ozone production rates. This could be a good paper, however it is currently lacking in several ways. First, the authors should describe the steps taken to determine the different terms in equation 1. Show a vertical profile of NOx, show how you determine zi, etc. This will make it easier for the reader to follow the authors' conclusions.

The writing style should be improved as well. This paper would be better if the authors introduced each section with some background information about what they are doing

and why. There are numerous grammatical mistakes throughout the paper. I have tried to correct some, listed below. Commas should separate introductory clauses in sentences. Often there are missing spaces between words and parentheses. Subscripts are sometimes missing.

Until these changes are made, I find it difficult to properly review it. Therefore, I recommend that major revisions are necessary.

Some other questions I had are as follows:

More explanation is needed for the boundary layer height (ABL). For these flights, what were the ABL heights determined from aircraft and from the model. What were they used in Equation 1?

p. 5, line 5, and Figure 2, define in what time period is this probability calculated?

p. 5, line 21, before using WRF for vertical mixing, how does the model compare with ABL heights?

p. 9, line 7, add units to 6x10ˆ6

Table 1 needs more information/description. Are the authors solving for F0? What are the estimates of zi on these days? Also, you should use the same notation for average scalar as in Equation (1).

p. 13, line 24, Please explain where this 59% number comes from

Section 4.1.1.5., explain what the Leighton ratio is before discussing the deviation of it and presenting a modified ratio

Supplemental Information, Figure 1: It appears the conversion had some formatting errors

Grammar

p. 2, line 2, add comma after "scalars" p. 3, line 9, change to "data tend to be" p. 9,

line 1, add comma after "budgets" p. 9, line 22, change "%50" to "50%" p. 10, line 2, change to "data were" p. 11, line 22, add a comma after "In their model for soil NOx" p. 12, line 13, start sentence with "It is . . ." p. 12, line 14, add comma after "satellite" p. 12, line 21, add comma after "urban air" p. 13, line 6, subscript the 2 in "NO2" p. 14, line 20, change "try and" to "try to", and subscript "NO2" p. 14, line 25, change "lose" to "loss", and subscript "NO2" p. 15, line 12, I'm confused by "14-6" p. 16, line 3, end sentence after "temperature" p. 16, line 16, the sentence beginning with "Marr" needs to be re-written p. 16, line 23, change "where" to "were" p. 18, lines 11 and 12, subscript the 3 in "O3" p. 20, line 16, change comma between "CH4" and "NOx" to "and" p. 21, line 9, subscripts for NOx and CH4 Figure 6 caption, there is something missing between "2." and "10.7"

---

## Author Comment (AC1) · 12 Mar 2019

Response to Referee #1 comments on "Photochemical Production of Ozone and Emissions of $NO_x$ and $CH_4$ in the San Joaquin Valley" published 19-Feb-19

*We thank the referee for their thorough reading of the manuscript, and address the individual comments below:*

General Comments:

1) There are numerous typos and grammatical errors. Some of these are listed be- low, but I would suggest a thorough proof-reading before resubmitting the manuscript. Some of the language is also vague and inappropriate for a journal paper (e.g. "stuck out", "more or less", "more and more", "a lot").

*Yes, thank you for pointing that out, we have double-checked the manuscript and removed the more imprecise colloquialisms.*

2) No details of the flights are presented, with the exception of two sentences in Section 3.3. Figure 1 gives some idea of the horizontal extent of the two flight campaigns, although it is difficult to see the EPA flight lines and it is impossible to distinguish individual flights. We are given some windows of time (but no actual flight durations) and an altitude range "from the surface up to 4 km". Presumably the surface measurements are at the start and end of each flight, unless there were multiple landings at different airports. Given the importance of the vertical coverage on the emission calculation, much more detail is required.

*We have expanded on the paragraph in Section 3.3 detailing the flight strategies to read, "The main data set we use here comes from six flights sponsored by the US EPA (labelled EPA in Figure 1) during the California Baseline Ozone Transport Study (CABOTS) that were conducted on the afternoons of 26-28 July and 4-6 August, 2016 from 1100 to 1500 PST spanning an approximate altitude range from near surface up to ~3 km. The aircraft flights consisted of 6-7 back and forth level and profiling legs of approximately 15 minutes duration (~60 km) up and back along the mean wind direction (the valley axis) in order to capture the horizontal advection and vertical gradients of the measured scalars. The flight domain focused on the region of the SJV between Fresno and Visalia with approximately two-thirds of the data collected below ~1 km . The flight days were selected in coordination with a crew from NOAA operating a Tunable Optical Profiler for Aerosol and Ozone (TOPAZ) lidar in Visalia, California (Langford et al., 2019). Periodically the plane would make deep vertical profiles from ~10 m to 3 km in addition to two or three other profiling legs in order to diagnosis the ABL top, its growth, and vertical profiles of the measured scalars."*

3) The description of the NOx processing (Section 4.1.1) lacks detail. Why one standard deviation? How much data are removed? If this all occurs in the late afternoon, why not just reject data from that time of day? And if fire smoke is entrained during that time period, why is this effect not discussed for $O_3$ and $CH_4$?

*We have clarified and justified the explanation of the method of removing the NO$_x$ spikes by stating, "NO$_x$ ABL data was filtered by eliminating data greater than one standard deviation above the mean before being analyzed in order to remove the skewness from the distributions induced by numerous spikes encountered in the late afternoons. Variations of this threshold from 1-3 standard deviations did not change the mean flight concentration by more than 2-3 percent so the exact threshold was not considered critical for our analysis. The data filtering was done to eliminate the spikes which were consistently encountered throughout the latter part of the flights, each lasting no more than a few minutes and uncorrelated with any other species measured (CO$_2$, CH$_4$, and O$_3$.) We conjecture that their source may have been something in the fire smoke entrained in the late afternoon ABL that caused a transient interference in the NO$_2$ photolytic chamber (they were not observed in the NO measurements.) Furthermore, as we discuss later in conjunction with Figure 5, the influence of the fires on NO$_x$ measured by the surface network (~1 ppbv) appears to be minimal relative to the clear signal enhancements in CO (~200 ppbv) and PM2.5 (~15 $\mu gm^{-3}$)."*

4) If emission rates are measured over the course of a few hours for 6 days, then presenting these values as month or yearly rates (i.e. tonnes/year) is not "averaging" or "converting" - It is extrapolating. For the most part this can be fixed by using the correct terminology. However, due to variations in emissions with time (some of which are discussed, such as the "weekend effect"), extrapolating these hourly time scale values to yearly values has an associated uncertainty that should be discussed.

*Of course, all measurements of limited duration need to be extrapolated in time; however, there are conventional units used in inventories which make for more convenient comparison. Here we use ppb/h, tonnes/day, and Mg/h (and Gg/yr) for the O3, NOx, and CH4 source strengths, so there is not all that much extrapolation in these units. The conversion from the four hour flight to the entire daily emissions of NOx is discussed extensively in Section 4.1.1.3, and we have added a statement in the methane emissions section (4.1.3) to acknowledge their seasonal temperature dependence:*

*"Because the airborne measurements of our study take place during six summer days, comparisons with annually averaged inventories should be done with caution as methane emissions from the dominant sources (dairies) are likely to be seasonally temperature dependent. For example, a recent study of two dairies in the SJV (Arndt et al., 2019) reported facility-wide winter emissions to be only 40-50% of those during summer sampling."*

5) The consideration of uncertainties is generally weak. For example, in Section 3.4 (page 8, line 16) it is simply stated that the approach is justified and that 20% in conservative without any reference to where that number comes from. In Section 4.4, two values (1 ppb, 50 ppb) are chosen "because the term is estimated by eye". For wind, 0.1 m/s is based on the "measurement capabilities" of an instrument which isn't named or referenced.

*At the core of this issue, is our belief that the estimation of compound measurement errors is important yet ambiguous guesswork, and we generally refrain from the presentation of precise quantitative determinations of errors lest they give the impression that they are fully understood. When it comes to applying a mesoscale area over which to integrate and interpret our measurements, the exact uncertainty of the domain is, frankly, uncertain. Nonetheless, we have added the following discussion to justify our crude estimate as conservative in Section 3.4:*

*"The average area of this polygon was 5,200 km$^2$ ($\sigma$=940 km$^2$). To estimate an uncertainty in this area, we consider the average advection distance of the mean ABL wind (~3 ms$^{-1}$) over the course of a large eddy turnover time (boundary layer height divided by convective velocity scale ~ 8 minutes = 650 m/1.35 ms$^{-1}$) and multiply this on either end of the domain by an average cross-valley dimension (70 km) to generate a 'spread' in the sampled ABL volume influenced by the surface flux field. Although this additional area represents less than 4% of the overall domain, we include a conservative 20% error in the error analysis for it."*

*And in Section 4.4 Error Analysis we have added:*

*"The delta term error was assigned to be 1.0 ppb for NO$_x$ and 50 ppb for methane based on variations in the data estimated by eye from inspection of the many vertical profiles."*

*"…and the horizontal wind at ABL height assigned an error of 0.1 ms$^{-1}$ based on the measurement capabilities of the instrument (Conley et al. 2014)."*

6) Correlation length is typically calculated at the 1/e value, not the "crossing of the zero-correlation line". This is primarily because small amount of noise in the correlation values can significantly change when the zero line is first crossed (Figure 12 demonstrates this effect). Smoothing the correlation or fitting an exponential decay to determine the value at 1/e gives a more accurate measure of the correlation length that isn't subject to the effects of noisy data.

*The statement was in error because we did in fact use the e-folding depth for the calculation. We have changed the statement in the paper.*

7) It is also not clear why correlation distances are important in the context of the study (expect to inform future satellite resolution values). What is the expected correlation length that would be associated with cities and traffic? Wouldn't this value depend on how far downwind from the source the measurement is (due to horizontal diffusion)? Why are potential temperature and water vapor compared? Do these values relate to the patchiness of land use and the separation of lakes and rivers? Why is this important?

*We are not exactly sure of the answers to these questions, because we have not found this detail discussed directly in the literature. However, we wanted to present these results because we feel it may spark a useful discussion in the future literature on regional air quality. Because*

*the measurements represent a sampled area of order 75X75 km, the average decorrelation length should include some 'average' of a total urban plume downwind. The main city centers of this region are approximately 5-10 km in linear dimension, whereas California Highway 99 runs straight through the ~75 km flight domain, so it is not obvious what a line source spatial scale should be exactly.*

*We have amended the discussion to be more direct about our speculation: "Temperature and water represent ABL scalars dominated by surface fluxes, so in principle their correlation lengths are related to the scale of heterogeneity in their surface sources (irrigated or fallow fields, plots of differing albedos, urban heat islands, etc.) In the case of ozone, photochemical production dominates in the afternoon requiring the mixing together of $NO_x$ and VOC emissions. The more spatially diffuse pattern of $CH_4$ and $NO_x$, comparable to that of ozone, may imply a preponderance of broad areal sources rather than localized emissions from cities (5-15 km) and/or highway traffic. This result for $NO_x$ calls to mind the findings from Russell et al. (2010) and Pusede and Cohen (2012) previously mentioned which show broad scale homogeneity for NOx concentrations in the SJV unlike in other regions where urban hotspots are more localized."*

Specific Comments:

Page 3, line 9-10: A claim like this is meaningless without defining "limited in duration", "overextended in sampling", and "altogether uncoordinated". If those terms can be defined a citation will also be needed to a substantial review paper that backs up this claim.

*This is an opinion of ours recalling one of the author's 20 years of experience with airborne atmospheric chemistry studies. As far as we can tell there is no author to date who has lamented the absence of such coordinated flight strategies, so there is nothing to cite.*

Page 5, line 17: For the editor – Is a citation of a manuscript in preparation accepted?

*Changed to: "The approximate residence time within this buffer layer was found to be about one week based on analysis of WRF model output which we plan to detail in a companion paper."*

Page 7, line 11: The aircraft doesn't measure "from the surface".

*In response to General Comment (2) above we have changed the wording to "near surface" and have explained about the occasional low-passes conducted at airports to sample within 5-10 m of the surface.*

Page 9, line 14: Using WRF parameterizations isn't a measurement.

*Changed wording.*

Page 20, line 12-13: This sentence is very confusing. (e.g. What is "a common time height stamp"?)

*changed to, "All data was selected to be within the time dependent ABL and corrected to a common time and height within the ABL to remove biasing from temporal and vertical trends before the autocorrelation was run."*

Minor comments/corrections:
Page 2, line 23. Sentence doesn't make sense.

*Split it into two sentences for clarity's sake: "In 1990, the San Joaquin Valley Air Quality Study (SJVAQS) was conducted. The largest study of its kind in the U.S. at the time, the SJVAQS targeted the complexities of the SJV at a time when it was considered the nation's second worst overall air quality problem (Lagarias and Sylte, 1991)."*

Page 3, line 14. Period should be outside bracket.

*That is not my understanding of conventional grammar.*

Page 5, line 3. What does "its" refer to?

*changed to "…reaching a maximum…"*

Page 5, line 11. Why is the air "unique"?

*changed to, "The air above the ABL in the SJV is unique in that it does not purely consist of background air from the free troposphere (FT) as in most flat terrain."*

Page 6, line 13. Should be "Vaisala". "Ozone" should start new sentence.

*Changed.*

Page 9, line 22. Should be "50%".

*Changed.*

Page 12, line 13. This is not a sentence.

*Fixed.*

---

## Editor Comment (EC1) · Robert McLaren (Editor) · 24 Apr 2019

Response to Referee #1 comments on "Photochemical Production of Ozone and Emissions of $NO_x$ and $CH_4$ in the San Joaquin Valley" published 19-Feb-19

*We thank the referee for their thorough reading of the manuscript, and address the individual comments below:*

General Comments:

1) There are numerous typos and grammatical errors. Some of these are listed below, but I would suggest a thorough proof-reading before resubmitting the manuscript. Some of the language is also vague and inappropriate for a journal paper (e.g. "stuck out", "more or less", "more and more", "a lot").

*Yes, thank you for pointing that out, we have double-chekced the manuscript and removed the more imprecise colloquialisms.*

2) No details of the flights are presented, with the exception of two sentences in Section 3.3. Figure 1 gives some idea of the horizontal extent of the two flight campaigns, although it is difficult to see the EPA flight lines and it is impossible to distinguish individual flights. We are given some windows of time (but no actual flight durations) and an altitude range "from the surface up to 4 km". Presumably the surface measurements are at the start and end of each flight, unless there were multiple landings at different airports. Given the importance of the vertical coverage on the emission calculation, much more detail is required.

*We have expanded the paragraph in Section 3.3 detailing the flight strategies and included a new supplemental figure (S1) to show a sample of the profile measurements. The new paragraph reads, "The main data set we use here comes from six flights sponsored by the US EPA (labelled EPA in Figure 1) during the California Baseline Ozone Transport Study (CABOTS) that were conducted on the afternoons of 26-28 July and 4-6 August, 2016 from 1100 to 1500 PST spanning an approximate altitude range from near the surface up to ~3 km (Figure S1). The aircraft flights consisted of 6-7 back and forth level and profiling legs of approximately 15 minutes duration (~60 km) up and back primarily along the mean wind direction (the valley axis) in order to capture the horizontal advection and vertical gradients of the measured scalars. The flight domain focused on the region of the SJV between Fresno and Visalia with approximately two-thirds of the data collected below ~1 km, and missed approaches executed at each airport in order to sample to within a few meters of the ground. The flight days were selected in coordination with a crew from NOAA operating a Tunable Optical Profiler for Aerosol and Ozone (TOPAZ) lidar in Visalia, California who have shown excellent correspondence between the aircraft and lidar (Langford et al., 2019). Periodically the plane would make deep vertical profiles from ~3 m to 3 km in addition to two or three other profiling legs in order to diagnosis the ABL top, its growth, and vertical profiles of the measured scalars."*

3) The description of the NOx processing (Section 4.1.1) lacks detail. Why one standard deviation? How much data are removed? If this all occurs in the late afternoon, why not just reject data from that time of day? And if fire smoke is entrained during that time period, why is this effect not discussed for $O_3$ and $CH_4$?

*We have clarified and justified the explanation of the method of removing the $NO_x$ spikes by stating, "$NO_x$ ABL data were filtered by eliminating data greater than one standard deviation above the mean before being analysed in order to remove the skewness from the distributions induced by numerous spikes encountered in the late afternoons. Variations of this threshold from 1-3 standard deviations did not change the mean flight concentration by more than 2-3 percent so the exact threshhold was not considered critical for our analysis. The data filtering was done to eliminate the spikes which were consistently encountered throughout the latter part of the flights, each lasting no more than a few minutes and uncorrelated with any other species measured ($CO_2$, $CH_4$, and $O_3$.) We conjecture that their source may have been something in the fire smoke entrained in the late afternoon ABL that caused a transient interference in the $NO_2$ photolytic chamber (they were not observed in the NO meausrements.) Furthermore, as we discuss later in conjunction with Figure 5, the influence of the fires on $NO_x$ measured by the surface network (~1 ppbv) appears to be minimal relative to the clear signal enhancements in CO (~200 ppbv) and PM2.5 (~15 $\mu gm^{-3}$)."*

4) If emission rates are measured over the course of a few hours for 6 days, then presenting these values as month or yearly rates (i.e. tonnes/year) is not "averaging" or "converting" - It is extrapolating. For the most part this can be fixed by using the correct terminology. However, due to variations in emissions with time (some of which are discussed, such as the "weekend effect"), extrapolating these hourly time scale values to yearly values has an associated uncertainty that should be discussed.

*Of course, all measurements of limited duration need to be extrapolated in time; however, there are conventional units used in inventories which make for more convenient comparison. Here we use ppb/h, tonnes/day, and Mg/h (and Gg/yr) for the $O_3$, $NO_x$, and $CH_4$ source strengths, so there is not all that much extrapolation in these units. The conversion from the four hour flight to the entire daily emissions of $NO_x$ is discussed extensively in Section 4.1.1.3, and we have added a statement in the methane emissions section (4.1.3) to acknowledge their seasonal temperature dependence:*

*"Because the airborne measurements of our study take place during six summer days, comparisons with annually averaged inventories should be done with caution as methane emissions from the dominant sources (dairies) are likely to be seasonally temperature dependent. For example, a recent study of two dairies in the SJV (Arndt et al., 2019) reported facility-wide winter emissions to be only 40-50% of those during summer sampling."*

5) The consideration of uncertainties is generally weak. For example, in Section 3.4 (page 8, line 16) it is simply stated that the approach is justified and that 20% in conservative without any

reference to where that number comes from. In Section 4.4, two values (1 ppb, 50 ppb) are chosen "because the term is estimated by eye". For wind, 0.1 m/s is based on the "measurement capabilities" of an instrument which isn't named or referenced.

*At the core of this issue is our belief that the estimation of compound measurement errors is important yet ambiguous guesswork, and we generally refrain from the presentation of precise quantitative determinations of errors lest they give the impression that they are fully understood. When it comes to applying a mesoscale area over which to integrate and interpret our measurements, the exact uncertainty of the domain is, frankly, uncertain. Nonetheless, we have added the following discussion to justify our crude estimate as conservative in Section 3.4:*

*"The average area of this polygon was 5,200 $km^2$ ($\sigma$=940 $km^2$). To estimate an uncertainty in this area, we consider the average advection distance of the mean ABL wind (~3 $ms^{-1}$) over the course of a large eddy turnover time (boundary layer height divided by convective velocity scale ~ 8 minutes = 650 m/1.35 $ms^{-1}$) and multiply this on either end of the domain by an average cross-valley dimension (70 km) to generate a 'spread' in the sampled ABL volume influenced by the surface flux field. Although this additional area represents less than 4% of the overall domain, we include a conservative 20% error in the error analysis for it."*

*And in Section 4.4 Error Analysis we have added:*

*"The delta term error was assigned to be 1.0 ppb for $NO_x$ and 50 ppb for methane based on variations in the data estimated by eye from inspection of the many vertical profiles."*

*"…and the horizontal wind at ABL height assigned an error of 0.1 $ms^{-1}$ based on the measurement capabilities of the instrument (Conley et al. 2014)."*

6) Correlation length is typically calculated at the 1/e value, not the "crossing of the zero-correlation line". This is primarily because small amount of noise in the corre- lation values can significantly change when the zero line is first crossed (Figure 12 demonstrates this effect). Smoothing the correlation or fitting an exponential decay to determine the value at 1/e gives a more accurate measure of the correlation length that isn't subject to the effects of noisy data.

*In the past, we have found that the various ways of defining the decorrelation length scale do not lead to materially different results. However, in reviewing the analysis it was found that we are, in fact, using a 1/e crossing value to determine the decorrelation length. The text now reads, "The length scale was selected as the first crossing of the 0.37 line (=1/e)."*

7) It is also not clear why correlation distances are important in the context of the study (expect to inform future satellite resolution values). What is the expected correlation length that would be associated with cities and traffic? Wouldn't this value depend on how far downwind from the source the measurement is (due to horizontal diffusion)? Why are potential temperature

and water vapor compared? Do these values relate to the patchiness of land use and the separation of lakes and rivers? Why is this important?

*We are not exactly sure of the answers to these questions, because we have not found this detail discusssed directly in the literature.  However, we wanted to present these results because we feel it may spark a useful discussion in the future literature on regional air quality. Because the measurements represent a sampled area of order 75X75 km, the average decorrelation length should include some 'average' of a total urban plume downwind. The main city centers of this region are approximately 5-10 km in linear dimension, whereas Caliofrnia Highway 99 runs straight through the ~75 km flight domain, so it is not obvious what a line source spatial scale should be exactly.*

*We have ammended the discussion to be more direct about our speculation: "Temperature and water represent ABL scalars dominated by surface fluxes, so in principle their correlation lengths are related to the scale of heterogeneity in their surface sources (irrigated or fallow fields, plots of differing albedos, urban heat islands, etc.) In the case of ozone, photochemical production dominates in the afternoon requiring the mixing together of $NO_x$ and VOC emissions. The more spatially diffuse pattern of $CH_4$  and $NO_x$, comparable to that of ozone, may imply a preponderance of broad areal sources rather than localized emissions from cities (5-15 km) and/or highway traffic. This result for $NO_x$ calls to mind the findings from Russell et al. (2010) and Pusede and Cohen (2012) previously mentioned which show broad scale homogeneity for $NO_x$ concentrations in the SJV unlike in other regions where urban hotspots are more localized."*

Specific Comments:

Page 3, line 9-10: A claim like this is meaningless without defining "limited in duration", "overextended in sampling", and "altogether uncoordinated". If those terms can be defined a citation will also be needed to a substantial review paper that backs up this claim.

*This is an opinion of ours recalling one of the author's 20 years of experience with airborne atmospheric chemistry studies. As far as we can tell there is no author to date who has lamented the absence of such coordinated flight strategies, so there is nothing to cite.*

Page 5, line 17: For the editor – Is a citation of a manuscript in preparation accepted?

*Changed to:  "The approximate residence time within this buffer layer was found to be about one week based on analysis of WRF model output which we plan to detail in a forthcoming paper."*

Page 7, line 11: The aircraft doesn't measure "from the surface".

*In response to General Comment (2) above we have changed the wording to "near surface" and have explained about the occasional low-passes conducted at airports to sample within 5-10 m of the surface.*

Page 9, line 14: Using WRF parameterizations isn't a measurement.

*Changed wording.*

Page 20, line 12-13: This sentence is very confusing. (e.g. What is "a common time height stamp"?)

*changed to, "All data was selected to be within the time dependent ABL and corrected to a common time and height within the ABL to remove biasing from temporal and vertical trends before the autocorrelation was run."*

Minor comments/corrections:
Page 2, line 23. Sentence doesn't make sense.

*Split it into two sentences for clarity's sake: "In 1990, the San Joaquin Valley Air Quality Study (SJVAQS) was conducted. The largest study of its kind in the U.S. at the time, the SJVAQS targeted the complexities of the SJV at a time when it was considered the nation's second worst overall air quality problem (Lagarias and Sylte, 1991)."*

Page 3, line 14. Period should be outside bracket.

*That is not my understanding of conventional grammar.*

Page 5, line 3. What does "its" refer to?

*changed to "…reaching a maximum…"*

Page 5, line 11. Why is the air "unique"?

*changed to, "The air above the ABL in the SJV is unique in that it does not purely consist of background air from the free troposphere (FT) as in most cases over flat terrain."*

Page 6, line 13. Should be "Vaisala". "Ozone" should start new sentence.

*Changed.*

Page 9, line 22. Should be "50%".

*Changed.*

Page 12, line 13. This is not a sentence.

*Fixed.*

---

## Editor Comment (EC2) · Robert McLaren (Editor) · 24 Apr 2019

Response to Referee #2 comments on "Photochemical Production of Ozone and Emissions of $NO_x$ and $CH_4$ in the San Joaquin Valley" published 19-Feb-19
Review

*We thank the referee for their thorough reading of the manuscript, and address the individual comments below:*

Trousdell et al. present data from flights over the San Joaquin Valley in California. They calculate $NO_x$ and $CH_4$ emission rates for this region, as well as photochemical ozone production rates. This could be a good paper, however it is currently lacking in several ways. First, the authors should describe the steps taken to determine the different terms in equation 1. Show a vertical profile of $NO_x$, show how you determine zi, etc. This will make it easier for the reader to follow the authors' conclusions.

*Because these methods have been repeated in several previous papers from our group [Trousdell et al., 2016; Conley et al., 2009; Faloona et al., 2009; Conley et al., 2011] and the scope of this work is already expansive, we have chosen to minimiaze the step by step elaboration of the scalar budgeting method. On p.8, l.10 we state, "For a more in-depth discussion of the airborne budgeting technique and specifics for the budgets of methane and ozone in the SJV see Trousdell et al. (2016)." But we have added an example profile in the supplementary materials (new Figure S1) and a more explicitly methodological review in the text stating, "Boundary layer heights were determined from each profile (approximately 8-12 per flight) based on the abrupt increase in potential temperature and drop in water vapor. The locations and time of each of these observations were then fit by a multilinear regression in time and the horizontal dimension to determine the ABL growth rates and gradients which go into the budget to determine the entrainment velocity (Trousdell et al., 2016). Taking all the airborne data observed below the derived (linear) time-dependent ABL depth we then perform the same multi-linear regression for all the scalars including potential temperature, water vapor, $O_3$, $NO_x$, and $CH_4$. Aligning the x-axis with the mean wind direction, U, the advection and temporal trend terms of Equation 1 are derived from the coefficients of the linear regression fit to the ABL $NO_x$ concentration field in time and horizontal direction (Conley et al., 2011)."*

The writing style should be improved as well. This paper would be better if the authors **introduced each section** with some background information about **what they are doing and why**. There are numerous grammatical mistakes throughout the paper. I have tried to correct some, listed below. Commas should separate introductory clauses in sentences. Often there are missing spaces between words and parentheses. Subscripts are sometimes missing.
Until these changes are made, I find it difficult to properly review it. Therefore, I recommend that major revisions are necessary.

*Because Section 4 is very detailed and has many subsections, we have introduced the introductory paragraph below to help guide the reader through the reasoning of this reticulate section:*

*"4  Results and Discussion*

*In the following section we present a variety of inferences gleaned from the three scalar budgets performed for $NO_x$ to derive regional surface emissions (4.1.1), and for $O_3$ to derive afternoon photochemical production rates (4.1.2) and see how that fits in to the overall diurnal budget of ozone (4.1.2.1), and for $CH_4$ to derive regional emissions (4.1.3). Because of the large discrepancy between our estimates of $NO_x$ emissions and that of the state inventory, we further explore possible reasons to explain the difference. The first is the hypothesis put forward by Almaraz et al. (2018) that there is a substantial source of NO from fertilized agricultural soils that is not accounted for in current state inventories (4.1.1.1). The second is the possibility that the Soberanes Fire in the mountains of the Coast Range approximately 200 km to the west may have influenced our $NO_x$ budget in the ABL around Fresno (4.1.1.2). The third explores the bias introduced by measuring only during the afternoon when $NO_x$ emissions are thought to be highest (4.1.1.3), and the fourth discusses the possibility of a chemical interference in the measurement of $NO_2$, which in our system relies on photolysis followed by the chemiluminescence measurement of NO (4.1.1.4). The interference hypothesis is further explored by calculating Leighton ratios (4.1.1.5) in order to determine if the observed $NO_2$:NO ratios appear consistent with the theoretical photostationary state between $O_3$, NO, and $NO_2$ expressed in the Leighton ratios. This latter point leads naturally to the discussion of our estimates of ozone photochemical production (4.1.2) because it, in principle, is related to deviations in the observed Leighton ratios. Next, we present the observed spatial patterns of these scalars in the ABL calculating their horizontal autocorrelation lengths (4.2) to potentially infer emissions heterogeneity, and then finally we discuss the way we estimate the errors (4.3) in all the derived values of this budgeting study."*

Some other questions I had are as follows:
More explanation is needed for the boundary layer height (ABL). For these flights, what were the ABL heights determined from aircraft and from the model. What were they used in Equation 1?

*We did not include these details in this manuscript because we are preparing another, companion paper that focuses strictly on the entrainment and ABL dynamics of the valley.  That work will present boundary layer heights as well as the observed growth and advection rates, and ultimately the inferred entrainment velocities used in the scalar budgeting in this work. We have included the average boundary layer heights for each flight in Table 1, and where we mention this companion work we have added the average values:*

*"In a future companion paper, along with the boundary layer heights, $z_i$, (650 $\pm$ 50 m) and entrainment velocities, $w_e$, (3.0 $\pm$ 1.8 cms$^{-1}$), we present the surface sensible heat fluxes for our flight region via two independent methods."*

p. 5, line 5, and Figure 2, define in what time period is this probability calculated?

*The data interval of 2006-2015 is now mentioned in the text and in the figure caption.*

p. 5, line 21, before using WRF for vertical mixing, how does the model compare with ABL heights?

*The WRF model predicts ABL depths that are approximately 30% larger than our observations. However, we plan to discuss and explain this in the aforementioned campanion work to be submitted to "Boundary Layer Meteorology" soon. The WRF results central to this study are the vertical velocities at the top of the observed ABL heights, which should not be directly linked to the ABL results of the model.*

p. 9, line 7, add units to 6x10^6.  Table 1 needs more information/description. Are the authors solving for F0? What are the estimates of zi on these days? Also, you should use the same notation for average scalar as in Equation (1).

*Done.*

p. 13, line 24, Please explain where this 59% number comes from

*Upon reviewing the weekend/weekday bias, we found that we had overestimated its effect. We have rewritten the section to make it more clear, and the conclusion is that our sampling bias (due to hour of day and day of week combined) may be 45% higher than a long-term average inventory value as explained in the text:*

*" Assuming an average decrease of $NO_x$ emissions on weekends to 0.73 the weekday rate, our average daily emission rate would be a factor of 1.04 (=(5.73/6.46)x(7/6)) higher than inventories, which average over 5 weekdays and 2 weekend days. Taken together, the timing of the flights relative to the inventory's average summer emission rate could lead to a positive bias in our measurements of 45% (=1.4x1.04)."*

Section 4.1.1.5., explain what the Leighton ratio is before discussing the deviation of it and presenting a modified ratio

*Done.*

Supplemental Information, Figure 1: It appears the conversion had some formatting errors

*Apologies, the supplement has been reproduced (with an additional figure requested above) and resubmitted.*

Grammar:
p. 2, line 2, add comma after "scalars" p. 3, line 9, change to "data tend to be" p. 9, line 1, add comma after "budgets" p. 9, line 22, change "%50" to "50%" p. 10, line 2, change to "data were" p. 11, line 22, add a comma after "In their model for soil NOx" p. 12, line 13, start sentence with "It is . . ." p. 12, line 14, add comma after "satellite" p. 12, line 21, add comma after "urban air" p. 13, line 6, subscript the 2 in "NO2" p. 14, line 20, change "try and" to "try to", and subscript "NO2" p. 14, line 25, change "lose" to "loss", and subscript "NO2".
p. 15, line 12, I'm confused by "14-6" p. 16, line 3, end sentence after "temperature" p. 16, line 16, the sentence beginning with "Marr" needs to be re-written.
p. 16, line 23, change "where" to "were" p. 18, lines 11 and 12, subscript the 3 in "O3" p. 20, line 16, change comma between "CH4" and "NOx" to "and" p. 21, line 9, subscripts for NOx and CH4 Figure 6 caption, there is something missing between "2." and "10.7"

*Ok, these changes were made.*

---

## Author Response (AR1)

Response to Referee #2 comments on "Photochemical Production of Ozone and Emissions of $NO_x$ and $CH_4$ in the San Joaquin Valley" published 19-Feb-19
Review

*We thank the referee for their thorough reading of the manuscript, and address the individual comments below:*

Trousdell et al. present data from flights over the San Joaquin Valley in California. They calculate $NO_x$ and $CH_4$ emission rates for this region, as well as photochemical ozone production rates. This could be a good paper, however it is currently lacking in several ways. First, the authors should describe the steps taken to determine the different terms in equation 1. Show a vertical profile of $NO_x$, show how you determine zi, etc. This will make it easier for the reader to follow the authors' conclusions.

*Because these methods have been repeated in several previous papers from our group [Trousdell et al., 2016; Conley et al., 2009; Faloona et al., 2009; Conley et al., 2011] and the scope of this work is already expansive, we have chosen to minimiaze the step by step elaboration of the scalar budgeting method. On p.8, l.10 we state, "For a more in-depth discussion of the airborne budgeting technique and specifics for the budgets of methane and ozone in the SJV see Trousdell et al. (2016)." But we have added an example profile in the supplementary materials (new Figure S1) and a more explicitly methodological review in the text stating, "Boundary layer heights were determined from each profile (approximately 8-12 per flight) based on the abrupt increase in potential temperature and drop in water vapor. The locations and time of each of these observations were then fit by a multilinear regression in time and the horizontal dimension to determine the ABL growth rates and gradients which go into the budget to determine the entrainment velocity (Trousdell et al., 2016). Taking all the airborne data observed below the derived (linear) time-dependent ABL depth we then perform the same multi-linear regression for all the scalars including potential temperature, water vapor, $O_3$, $NO_x$, and $CH_4$. Aligning the x-axis with the mean wind direction, U, the advection and temporal trend terms of Equation 1 are derived from the coefficients of the linear regression fit to the ABL $NO_x$ concentration field in time and horizontal direction (Conley et al., 2011)."*

The writing style should be improved as well. This paper would be better if the authors ***introduced each section*** with some background information about ***what they are doing and why***. There are numerous grammatical mistakes throughout the paper. I have tried to correct some, listed below. Commas should separate introductory clauses in sentences. Often there are missing spaces between words and parentheses. Subscripts are sometimes missing.
Until these changes are made, I find it difficult to properly review it. Therefore, I recommend that major revisions are necessary.

*Because Section 4 is very detailed and has many subsections, we have introduced the introductory paragraph below to help guide the reader through the reasoning of this reticulate section:*

*"4  Results and Discussion*

*In the following section we present a variety of inferences gleaned from the three scalar budgets performed for $NO_x$ to derive regional surface emissions (4.1.1), and for $O_3$ to derive afternoon photochemical production rates (4.1.2) and see how that fits in to the overall diurnal budget of ozone (4.1.2.1), and for $CH_4$ to derive regional emissions (4.1.3). Because of the large discrepancy between our estimates of $NO_x$ emissions and that of the state inventory, we further explore possible reasons to explain the difference. The first is the hypothesis put forward by Almaraz et al. (2018) that there is a substantial source of NO from fertilized agricultural soils that is not accounted for in current state inventories (4.1.1.1). The second is the possibility that the Soberanes Fire in the mountains of the Coast Range approximately 200 km to the west may have influenced our $NO_x$ budget in the ABL around Fresno (4.1.1.2). The third explores the bias introduced by measuring only during the afternoon when $NO_x$ emissions are thought to be highest (4.1.1.3), and the fourth discusses the possibility of a chemical interference in the measurement of $NO_2$, which in our system relies on photolysis followed by the chemiluminescence measurement of NO (4.1.1.4). The interference hypothesis is further explored by calculating Leighton ratios (4.1.1.5) in order to determine if the observed $NO_2$:NO ratios appear consistent with the theoretical photostationary state between $O_3$, NO, and $NO_2$ expressed in the Leighton ratios. This latter point leads naturally to the discussion of our estimates of ozone photochemical production (4.1.2) because it, in principle, is related to deviations in the observed Leighton ratios. Next, we present the observed spatial patterns of these scalars in the ABL calculating their horizontal autocorrelation lengths (4.2) to potentially infer emissions heterogeneity, and then finally we discuss the way we estimate the errors (4.3) in all the derived values of this budgeting study."*

Some other questions I had are as follows:
More explanation is needed for the boundary layer height (ABL). For these flights, what were the ABL heights determined from aircraft and from the model. What were they used in Equation 1?

*We did not include these details in this manuscript because we are preparing another, companion paper that focuses strictly on the entrainment and ABL dynamics of the valley.  That work will present boundary layer heights as well as the observed growth and advection rates, and ultimately the inferred entrainment velocities used in the scalar budgeting in this work. We have included the average boundary layer heights for each flight in Table 1, and where we mention this companion work we have added the average values:*

*"In a future companion paper, along with the boundary layer heights, $z_i$, (650 $\pm$ 50 m) and entrainment velocities, $w_e$, (3.0 $\pm$ 1.8 cms$^{-1}$), we present the surface sensible heat fluxes for our flight region via two independent methods."*

p. 5, line 5, and Figure 2, define in what time period is this probability calculated?

*The data interval of 2006-2015 is now mentioned in the text and in the figure caption.*

p. 5, line 21, before using WRF for vertical mixing, how does the model compare with ABL heights?

*The WRF model predicts ABL depths that are approximately 30% larger than our observations. However, we plan to discuss and explain this in the aforementioned campanion work to be submitted to "Boundary Layer Meteorology" soon. The WRF results central to this study are the vertical velocities at the top of the observed ABL heights, which should not be directly linked to the ABL results of the model.*

p. 9, line 7, add units to 6x10ˆ6.  Table 1 needs more information/description. Are the authors solving for F0? What are the estimates of zi on these days? Also, you should use the same notation for average scalar as in Equation (1).

*Done.*

p. 13, line 24, Please explain where this 59% number comes from

*Upon reviewing the weekend/weekday bias, we found that we had overestimated its effect. We have rewritten the section to make it more clear, and the conclusion is that our sampling bias (due to hour of day and day of week combined) may be 45% higher than a long-term average inventory value as explained in the text:*

*" Assuming an average decrease of $NO_x$ emissions on weekends to 0.73 the weekday rate, our average daily emission rate would be a factor of 1.04 (=(5.73/6.46)x(7/6)) higher than inventories, which average over 5 weekdays and 2 weekend days. Taken together, the timing of the flights relative to the inventory's average summer emission rate could lead to a positive bias in our measurements of 45% (=1.4x1.04)."*

Section 4.1.1.5., explain what the Leighton ratio is before discussing the deviation of it and presenting a modified ratio

*Done.*

Supplemental Information, Figure 1: It appears the conversion had some formatting errors

*Apologies, the supplement has been reproduced (with an additional figure requested above) and resubmitted.*

Grammar:
p. 2, line 2, add comma after "scalars" p. 3, line 9, change to "data tend to be" p. 9, line 1, add comma after "budgets" p. 9, line 22, change "%50" to "50%" p. 10, line 2, change to "data were" p. 11, line 22, add a comma after "In their model for soil NOx" p. 12, line 13, start sentence with "It is . . ." p. 12, line 14, add comma after "satellite" p. 12, line 21, add comma after "urban air" p. 13, line 6, subscript the 2 in "NO2" p. 14, line 20, change "try and" to "try to", and subscript "NO2" p. 14, line 25, change "lose" to "loss", and subscript "NO2".
p. 15, line 12, I'm confused by "14-6" p. 16, line 3, end sentence after "temperature" p. 16, line 16, the sentence beginning with "Marr" needs to be re-written.
p. 16, line 23, change "where" to "were" p. 18, lines 11 and 12, subscript the 3 in "O3" p. 20, line 16, change comma between "CH4" and "NOx" to "and" p. 21, line 9, subscripts for NOx and CH4 Figure 6 caption, there is something missing between "2." and "10.7"

*Ok, these changes were made.*

Response to Referee #1 comments on "Photochemical Production of Ozone and Emissions of $NO_x$ and $CH_4$ in the San Joaquin Valley" published 19-Feb-19

*We thank the referee for their thorough reading of the manuscript, and address the individual comments below:*

General Comments:

1) There are numerous typos and grammatical errors. Some of these are listed below, but I would suggest a thorough proof-reading before resubmitting the manuscript. Some of the language is also vague and inappropriate for a journal paper (e.g. "stuck out", "more or less", "more and more", "a lot").

*Yes, thank you for pointing that out, we have double-chekced the manuscript and removed the more imprecise colloquialisms.*

2) No details of the flights are presented, with the exception of two sentences in Section 3.3. Figure 1 gives some idea of the horizontal extent of the two flight campaigns, although it is difficult to see the EPA flight lines and it is impossible to distinguish individual flights. We are given some windows of time (but no actual flight durations) and an altitude range "from the surface up to 4 km". Presumably the surface measurements are at the start and end of each flight, unless there were multiple landings at different airports. Given the importance of the vertical coverage on the emission calculation, much more detail is required.

*We have expanded the paragraph in Section 3.3 detailing the flight strategies and included a new supplemental figure (S1) to show a sample of the profile measurements. The new paragraph reads, "The main data set we use here comes from six flights sponsored by the US EPA (labelled EPA in Figure 1) during the California Baseline Ozone Transport Study (CABOTS) that were conducted on the afternoons of 26-28 July and 4-6 August, 2016 from 1100 to 1500 PST spanning an approximate altitude range from near the surface up to ~3 km (Figure S1). The aircraft flights consisted of 6-7 back and forth level and profiling legs of approximately 15 minutes duration (~60 km) up and back primarily along the mean wind direction (the valley axis) in order to capture the horizontal advection and vertical gradients of the measured scalars. The flight domain focused on the region of the SJV between Fresno and Visalia with approximately two-thirds of the data collected below ~1 km, and missed approaches executed at each airport in order to sample to within a few meters of the ground. The flight days were selected in coordination with a crew from NOAA operating a Tunable Optical Profiler for Aerosol and Ozone (TOPAZ) lidar in Visalia, California who have shown excellent correspondence between the aircraft and lidar (Langford et al., 2019). Periodically the plane would make deep vertical profiles from ~3 m to 3 km in addition to two or three other profiling legs in order to diagnosis the ABL top, its growth, and vertical profiles of the measured scalars."*

3) The description of the NOx processing (Section 4.1.1) lacks detail. Why one standard deviation? How much data are removed? If this all occurs in the late afternoon, why not just reject data from that time of day? And if fire smoke is entrained during that time period, why is this effect not discussed for $O_3$ and $CH_4$?

*We have clarified and justified the explanation of the method of removing the $NO_x$ spikes by stating, "$NO_x$ ABL data were filtered by eliminating data greater than one standard deviation above the mean before being analysed in order to remove the skewness from the distributions induced by numerous spikes encountered in the late afternoons. Variations of this threshold from 1-3 standard deviations did not change the mean flight concentration by more than 2-3 percent so the exact threshhold was not considered critical for our analysis. The data filtering was done to eliminate the spikes which were consistently encountered throughout the latter part of the flights, each lasting no more than a few minutes and uncorrelated with any other species measured ($CO_2$, $CH_4$, and $O_3$.) We conjecture that their source may have been something in the fire smoke entrained in the late afternoon ABL that caused a transient interference in the $NO_2$ photolytic chamber (they were not observed in the NO meausrements.) Furthermore, as we discuss later in conjunction with Figure 5, the influence of the fires on $NO_x$ measured by the surface network (~1 ppbv) appears to be minimal relative to the clear signal enhancements in CO (~200 ppbv) and PM2.5 (~15 $\mu gm^{-3}$)."*

4) If emission rates are measured over the course of a few hours for 6 days, then presenting these values as month or yearly rates (i.e. tonnes/year) is not "averaging" or "converting" - It is extrapolating. For the most part this can be fixed by using the correct terminology. However, due to variations in emissions with time (some of which are discussed, such as the "weekend effect"), extrapolating these hourly time scale values to yearly values has an associated uncertainty that should be discussed.

*Of course, all measurements of limited duration need to be extrapolated in time; however, there are conventional units used in inventories which make for more convenient comparison. Here we use ppb/h, tonnes/day, and Mg/h (and Gg/yr) for the $O_3$, $NO_x$, and $CH_4$ source strengths, so there is not all that much extrapolation in these units. The conversion from the four hour flight to the entire daily emissions of $NO_x$ is discussed extensively in Section 4.1.1.3, and we have added a statement in the methane emissions section (4.1.3) to acknowledge their seasonal temperature dependence:*

*"Because the airborne measurements of our study take place during six summer days, comparisons with annually averaged inventories should be done with caution as methane emissions from the dominant sources (dairies) are likely to be seasonally temperature dependent. For example, a recent study of two dairies in the SJV (Arndt et al., 2019) reported facility-wide winter emissions to be only 40-50% of those during summer sampling."*

5) The consideration of uncertainties is generally weak. For example, in Section 3.4 (page 8, line 16) it is simply stated that the approach is justified and that 20% in conservative without any

reference to where that number comes from. In Section 4.4, two values (1 ppb, 50 ppb) are chosen "because the term is estimated by eye". For wind, 0.1 m/s is based on the "measurement capabilities" of an instrument which isn't named or referenced.

*At the core of this issue is our belief that the estimation of compound measurement errors is important yet ambiguous guesswork, and we generally refrain from the presentation of precise quantitative determinations of errors lest they give the impression that they are fully understood. When it comes to applying a mesoscale area over which to integrate and interpret our measurements, the exact uncertainty of the domain is, frankly, uncertain. Nonetheless, we have added the following discussion to justify our crude estimate as conservative in Section 3.4:*

*"The average area of this polygon was 5,200 km$^2$ ($\sigma$=940 km$^2$). To estimate an uncertainty in this area, we consider the average advection distance of the mean ABL wind (~3 ms$^{-1}$) over the course of a large eddy turnover time (boundary layer height divided by convective velocity scale ~ 8 minutes = 650 m/1.35 ms$^{-1}$) and multiply this on either end of the domain by an average cross-valley dimension (70 km) to generate a 'spread' in the sampled ABL volume influenced by the surface flux field. Although this additional area represents less than 4% of the overall domain, we include a conservative 20% error in the error analysis for it."*

*And in Section 4.4 Error Analysis we have added:*

*"The delta term error was assigned to be 1.0 ppb for NO$_X$ and 50 ppb for methane based on variations in the data estimated by eye from inspection of the many vertical profiles."*

*"…and the horizontal wind at ABL height assigned an error of 0.1 ms$^{-1}$ based on the measurement capabilities of the instrument (Conley et al. 2014)."*

6) Correlation length is typically calculated at the 1/e value, not the "crossing of the zero-correlation line". This is primarily because small amount of noise in the corre- lation values can significantly change when the zero line is first crossed (Figure 12 demonstrates this effect). Smoothing the correlation or fitting an exponential decay to determine the value at 1/e gives a more accurate measure of the correlation length that isn't subject to the effects of noisy data.

*In the past, we have found that the various ways of defining the decorrelation length scale do not lead to materially different results. However, in reviewing the analysis it was found that we are, in fact, using a 1/e crossing value to determine the decorrelation length. The text now reads, "The length scale was selected as the first crossing of the 0.37 line (=1/e)."*

7) It is also not clear why correlation distances are important in the context of the study (expect to inform future satellite resolution values). What is the expected correlation length that would be associated with cities and traffic? Wouldn't this value depend on how far downwind from the source the measurement is (due to horizontal diffusion)? Why are potential temperature

and water vapor compared? Do these values relate to the patchiness of land use and the separation of lakes and rivers? Why is this important?

*We are not exactly sure of the answers to these questions, because we have not found this detail discusssed directly in the literature.  However, we wanted to present these results because we feel it may spark a useful discussion in the future literature on regional air quality. Because the measurements represent a sampled area of order 75X75 km, the average decorrelation length should include some 'average' of a total urban plume downwind. The main city centers of this region are approximately 5-10 km in linear dimension, whereas Caliofrnia Highway 99 runs straight through the ~75 km flight domain, so it is not obvious what a line source spatial scale should be exactly.*

*We have ammended the discussion to be more direct about our speculation: "Temperature and water represent ABL scalars dominated by surface fluxes, so in principle their correlation lengths are related to the scale of heterogeneity in their surface sources (irrigated or fallow fields, plots of differing albedos, urban heat islands, etc.) In the case of ozone, photochemical production dominates in the afternoon requiring the mixing together of $NO_x$ and VOC emissions. The more spatially diffuse pattern of $CH_4$ and $NO_x$, comparable to that of ozone, may imply a preponderance of broad areal sources rather than localized emissions from cities (5-15 km) and/or highway traffic. This result for $NO_x$ calls to mind the findings from Russell et al. (2010) and Pusede and Cohen (2012) previously mentioned which show broad scale homogeneity for $NO_x$ concentrations in the SJV unlike in other regions where urban hotspots are more localized."*

Specific Comments:

Page 3, line 9-10: A claim like this is meaningless without defining "limited in duration", "overextended in sampling", and "altogether uncoordinated". If those terms can be defined a citation will also be needed to a substantial review paper that backs up this claim.

*This is an opinion of ours recalling one of the author's 20 years of experience with airborne atmospheric chemistry studies. As far as we can tell there is no author to date who has lamented the absence of such coordinated flight strategies, so there is nothing to cite.*

Page 5, line 17: For the editor – Is a citation of a manuscript in preparation accepted?

*Changed to:  "The approximate residence time within this buffer layer was found to be about one week based on analysis of WRF model output which we plan to detail in a forthcoming paper."*

Page 7, line 11: The aircraft doesn't measure "from the surface".

*In response to General Comment (2) above we have changed the wording to "near surface" and have explained about the occasional low-passes conducted at airports to sample within 5-10 m of the surface.*

Page 9, line 14: Using WRF parameterizations isn't a measurement.

*Changed wording.*

Page 20, line 12-13: This sentence is very confusing. (e.g. What is "a common time height stamp"?)

[revised manuscript text omitted]

---

## Author Response (AR2)

Review Comments Referee #2

Comments made to manuscript-version5

The authors have answered most of the reviewers' questions adequately. I have a few more suggestions for this paper before I think it is ready for publication.

I think the Uncertainties section should come earlier in the paper, before the final estimates of emissions are presented. Are the uncertainties, say for the $NO_x$ emissions, weighted by the daily uncertainties? In other words, was 182 ± 324 tonnes/d treated the same as 301 ± 166 tonnes/d? I also think the uncertainties should be clearly stated in this section, rather than scattered throughout. I found the boundary layer estimate of 650 ± 50 thrown in a section where it didn't really fit. A better place for it would be the Uncertainty section. Further, the authors state an uncertainty in the wind measurement of 0.1 m/s. I realize many of the coauthors wrote the original wind measurement paper, but my reading of the stated uncertainty in Conley et al. (2014) as < 0.2 m/s. Did the Mooney aircraft fly an L-pattern during this SJV project? Are the GPS measurements even accurate to 10 cm?

*We have moved Section 4.2 Error Analysis up to the budgeting methods section, 3.4.2. to more directly address these questions in the context of the methodology.*

*The overall uncertainties expressed as the standard deviation of the mean, are not weighted by each flight's uncertainty estimate.*

*When reporting the average boundary layer height, 650 ± 50 m, it was not meant as a source of uncertainty, because it's not that significant of one, but simply to give a physical idea of the depths observed (and their day to day regularity, here ± 50 m is the standard deviation of each day's average. Nevertheless, the concern about these parameters being elided due to a companion paper is acknowledged and we have added a table of the boundary layer parameters into the supplementary materials (see below).*

*We agree that a better wind measurement error estimate would be 0.2 m/s in keeping with the reference and have changed that. We have experiential reason to believe that the wind errors are less than that, but there is no reason to exclude that from our error estimates here.*

*The Mooney aircraft did not routinely fly wind calibration patterns during this SJV project, but it did so in between the deployments due to other efforts to quantify small-scale site emissions. The calibration coefficients are most often found not to change significantly during these checks and are updated in the data analysis codes when any wind system configuration change occurs. Finally, the individual GPS measurements are not likely to be accurate to within 10 cm, but the utility in the differential GPS system lies in each antenna's offset approximately canceling. The error reported in Conley et al. (2014) of <0.2 m/s is an aggregate of empirical testing against other wind measurements, not bottom-up estimates of the system components.*

If this paper is going to offload much of the boundary layer analysis to the companion paper, the companion paper should be provided along with this one.

*As mentioned above, we have added a supplemental Table S1 to detail the observed boundary layer parameters from this study in order to decouple this work from the "future companion paper" as much as possible. Due to many modeling improvements the exact details (e.g. entrainment velocities) of the upcoming paper will be slightly different from the values used here (but are absolutely within our estimated errors) and we feel it is important to document such differences, albeit minor, in print.*

p. 6, line 20, the authors should add the model number of the Picarro, like they did for the Eco Physics instrument.

*Done.*

In section 3.2, Aircraft Instrumentation, please add the estimated uncertainties to these measurements.

*Done.*

Section 4.1.1.2, how does the possible influence of the Soberanes fire compare with your estimate of the footprint? If the fire did affect the measurements, are your footprints too small? And if the fire didn't affect the measurements, why spend so much time discussing the possible interferences?

*On some days the overlap appeared to be quite expansive (e.g., https://earthobservatory.nasa.gov/images/88440/soberanes-fire-california) however the fire effluent was mostly in the layer above the ABL until afternoon convective development brought some of it into the ABL.*

*I do not understand the question, "are your footprints too small"? The footprints are determined by the aircraft paths from each flight, independent of the state of the fire effluent. We spend time discussing this possibility because it was an unanticipated environmental condition of the experiment that should be considered when trying to perform the budgets of these trace gases, which are all known to be influenced by wildfire emissions. We conclude that the **direct** emissions of fire $NO_x$ likely did not make a significant impact on our budget, but other reactive nitrogen species may have contributed to the analytic artifact in our $NO_x$ measurement that we also discuss. Our $NO_x$ emission estimates have already been reported in the literature (Almaraz et al., 2018), but with scant justification of the methods. In the present work we are attempting to systematically address as many potential problems with the analysis as possible and then make a probabilistic argument that our estimates are still significantly larger than inventories because of a missing soil source.*

p. 21, line 10, what units are 3.6 and 2.4? Are these scaling factors?

*They are ratios of top-down airborne estimates to emission inventory estimates as reported in Trousdell et al. (2016). We have adjusted the wording to make that clearer.*

Maasakkers et al. (2016) should account for the seasonality of CH4 emissions. How do CH4 emissions in the SJV compare to that work?

*We do not directly account for the seasonality of $CH_4$ emissions in this work, nor does Cui et al. (2017). We do, however, speculate that the seasonality of dairy emissions as reported in Arndt et al. (2018) could bring the estimates in much closer alignment.*

*We have decided **not** to include a comparison to the different, national inventory of Maasakkers et al. (2016) for a few reasons. First, their inventory shows a very low correlation to that of CALGEM, which has been extensively studied and optimized for California, for their livestock category ($r^2 \sim 0.21$). Second, the comparison is particularly poor (their Figure 6) in the exact region of this analysis. They state, "The largest differences are from livestock emissions, as CALGEM uses more local data to distribute these emissions within the large California counties." Further, their accounting for annual variations are only applied to manure management and they admit, "Livestock emissions also vary subanually as a function of varying herd size, and management practices but those effects are not included in our inventory."*

Figure 2, you could also refer to Figure 1 for the locations of the cities.

*OK.*

Supplement, p. 3, is this a correction factor that is multiplied by the measured NOx? If so, please add the additional information, e.g., NOx(corrected) = (factor)*NOx(measured)

*Corrected.*

Grammar Suggestions
p. 2, line 3, change "seem" to "seems"
*check.*
p. 4, line 7, add a comma after "chemistry"
*check.*
p. 5, line 11-12, set the clause "coupled … floor" off with commas, and remove the other commas in this sentence
*check.*
p. 6, line 16 and throughout, "data" should be plural
*check.*
p. 7, line 21, I'm confused by this statement. The NOAA crew corresponded well with the Mooney crew? Or the ozone measurements agreed well?

*Changed to, "who have shown excellent agreement between the ozone data collected by the aircraft and lidar (Langford et al., 2019)."*

p. 7, line 23, change "diagnosis" to "diagnose"

*check.*

p. 12, line 4, "Results from …" is in boldface

*check.*

p. 16, line 24, change "in" to "by"

*check.*

p. 20, line 5, "Data are"

*check.*

p. 20, line 13, I think d(Cox) should be d(Ox)?

*check.*

Supplement, p. 2, line 1, subscript the 2 in NO2

*check.*